# HiMAE: Hierarchical Masked Autoencoders Discover Resolution-Specific Structure in Wearable Time Series

**Simon A. Lee**[1,2,*]**, Cyrus Tanade**[1]**, Hao Zhou**[1]**, Juhyeon Lee**[1]**, Megha Thukral**[1]**, Md Sazzad, Hissain Khan**[1]**, Keum San Chun**[1]**, Baiying Lu**[1]**, Migyeong Gwak**[1]**, Mehrab Morshed**[1]**, Viswam Nathan**[1]**, Md Mahbubur Rahman**[1]**, Li Zhu**[1]**, Subramaniam Venkatraman**[1]**, Sharanya Arcot Desai**[1]**,**
[1]Digital Health Team, Samsung Research America
[2]Department of Computational Medicine, UCLA
* Work completed during AI Residency
simonlee711@g.ucla.edu

## Abstract

Wearable sensors provide abundant physiological time series observations, yet the resolution at which we should extract features for downstream tasks remain unclear. We hypothesize that temporal resolution is a fundamental axis of representation learning, with different clinical and behavioral outcomes relying on features at distinct scales. To test this resolution hypothesis, we introduce HiMAE (Hierarchical Masked Autoencoder), a self-supervised framework that combines masked autoencoding with a hierarchical convolutional encoder–decoder. HiMAE produces multi-resolution embeddings across its intermediate layers that enable systematic evaluation of which temporal scales carry predictive signal, transforming resolution from a hyperparameter into a probe for interpretability. Across classification and generative benchmarks, HiMAE consistently outperforms state-of-the-art foundation models that collapse scale, while being orders of magnitude smaller. Due to the convolution based design choices behind HiMAE, the model is also compact enough to run entirely on-device, achieving sub-millisecond inference on smartwatch-class CPUs for true edge inference. Together, these contributions position HiMAE as both an efficient self supervised learning method and a discovery tool for understanding how time resolution contributes to downstream task alignment.

## 1 Introduction

Wearable sensors have emerged as a primary modality for continuous health monitoring, providing access to rich physiological and behavioral signals in free-living settings (Erturk et al., 2025). Despite their ubiquity, the utility of wearable signals for machine learning in healthcare remains poorly understood. Unlike images (Dosovitskiy et al., 2021; Simonyan et al., 2014; Zhou et al., 2015; Petsiuk et al., 2018) or text (Brown et al., 2020; Li et al., 2016; Sundararajan et al., 2017; Arras et al., 2017), physiological time series rarely admit obvious visual cues that map cleanly to clinical outcomes, leaving open fundamental questions about which features carry predictive value. A particularly unresolved issue concerns temporal resolution: should models operate at a single universal resolution, or do different health outcomes depend on resolution-specific structure? Clinically actionable events can arise on second-level timescales, requiring representations that both capture fine-grained temporal patterns and support real-time inference under the computational constraints of wearable devices. We hypothesize that resolution is not a nuisance parameter but a fundamental axis of physiological representation learning. We refer to this as the *resolution hypothesis*, which posits that temporal granularity governs predictive performance in clinical and behavioral tasks. In this framing, "resolution" denotes the effective temporal context over which representations are formed—from fine-scale waveform morphology to coarse-scale dynamics spanning the whole sequence.

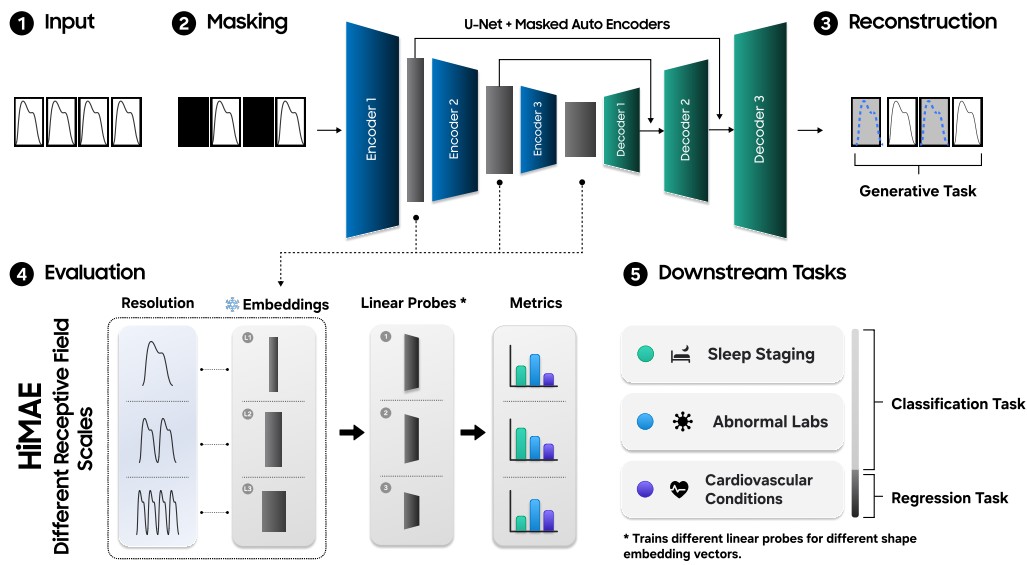

Figure 1: **HiMAE pre-training and evaluation pipeline.** (1) Physiological sequences are split into temporal patches. (2) Selected patches are masked randomly or contiguously. (3) A U-Net–style CNN encoder–decoder reconstructs missing values, with loss applied only to masked regions. (4) Multi-resolution embeddings feed linear probes for classification and regression benchmarking. (5) Three categorized task-lists are evaluated.

From an algorithmic perspective, much of the field defaults to transformer-based architectures (Vaswani et al., 2017), implicitly assuming that flexibility and capacity outweigh inductive bias. Yet wearable signals, while long in sequence length, are often generated by a few latent processes driven by biological mechanisms and captured through only a handful of sensor modalities. In this sense they are low-dimensional and highly structured. This raises the possibility that transformers may not only overfit but also obscure resolution-specific structure, rather than expose it. By contrast, hierarchical convolutional biases offer a natural mechanism for aligning architectures with the resolution hypothesis, capturing both local detail and long-range dependencies in a principled way. This motivates a re-examination of architectural design choices for self-supervised learning (SSL) on raw physiological time series.

In this work, we address these challenges by introducing *HiMAE* (Hierarchical Masked Autoencoder), a self-supervised pretraining framework for wearable time series that directly operationalizes the *resolution hypothesis* (Figure 1). HiMAE combines the masked autoencoding paradigm with 1D physiological signals by coupling patch-masking objectives (Wang et al., 2023) with a U-Net–inspired encoder–decoder (Ronneberger et al., 2015). Crucially, HiMAE produces multi-resolution embeddings, with each level of the hierarchy corresponding to a distinct temporal granularity. This design enables systematic interrogation of which resolutions carry predictive signal, while simultaneously yielding lightweight, efficient representations. Beyond its architectural advantages, HiMAE allows us to benchmark the resolution hypothesis across 14 classification. Our results reveal resolution-specific structure in wearable signals that is not readily identifiable by human experts, offering new insights into both representation learning and the interpretability of physiological time series in the time domain.

## 2 RELATED WORK

**Self-Supervised Pretraining Objectives for Wearable Signals** Wearable devices equipped with photoplethysmography (PPG), electrocardiography (ECG), and accelerometry generate long, multi-channel time series encoding diverse physiological and behavioral phenomena, including cardiovascular dynamics (Castaneda et al., 2018), activity patterns (Yuan et al., 2024; Xu et al., 2025), sleep cycles (Li et al., 2021; Thapa et al., 2024; Logacjov et al., 2025), and other latent processes. These data streams are abundant, and predominantly unlabeled, making them well suited for large-scale

self-supervised learning (Kaplan et al., 2020; Bommasani et al., 2021; Zhou et al., 2024; Liang et al., 2024).

SSL has become the dominant paradigm for wearable time-series representation learning, given the scarcity of labeled data and the ubiquity of unlabeled signals in free-living settings. Among SSL strategies, masked autoencoding has emerged as a central approach, inspired by its success in vision (He et al., 2022; Vaid et al., 2023) and language modeling (Devlin et al., 2019). The method randomly occludes patches of the signal and tasks the model with reconstructing them, encouraging representations that capture latent physiological structure and temporal regularities (Zhang et al., 2022a; Kong et al., 2023; Thukral et al., 2026; Lee et al.). Recent large-scale efforts, most notably Google's LSM series (Narayanswamy et al., 2024; Xu et al., 2025), rely heavily on masked autoencoding, establishing it as a pretraining standard for multi-modal wearable datasets. Yet despite its effectiveness for local pattern recovery, vanilla masked autoencoding often struggles to capture multi-resolution features unless coupled with explicitly hierarchical architectures.

In parallel, contrastive learning enforces invariance by pulling semantically similar samples together in latent space while pushing dissimilar ones apart (Schmitt & Kuljanin, 2008; Jaiswal et al., 2020). The central challenge for wearables is defining positive and negative pairs without labels. One solution is participant-level contrastive training, where samples from the same individual are positives and samples from different individuals are negatives, an approach adopted in Apple's ECG and PPG foundation models (Abbaspourazad et al., 2023) and closely related to the SimCLR framework (Chen et al., 2020b). Other domain-specific innovations define pairs through physiological priors: PaPaGei leverages PPG morphology (Pillai et al., 2024), while SleepFM extends the paradigm across EEG, ECG, and EMG to enforce cross-modal consistency (Thapa et al., 2024). Additional embedding-level regularizers, such as differential entropy constraints (Jing et al., 2021; Abbaspourazad et al., 2023), further enrich learned representations. However, contrastive methods are highly sensitive to augmentation heuristics (which are rarely physilogically meaningful), computationally intensive, and limited in interpretability, providing little insight into which temporal structures are preserved.

HiMAE departs from both flat masked and contrastive approaches in two ways. First, instead of relying on a single-scale reconstruction or augmentation heuristics, HiMAE couples masked autoencoding with a hierarchical encoder–decoder that integrates information across resolutions, treating temporal scale as an explicit dimension of representation. Second, by extracting embeddings at multiple scales and probing them independently, HiMAE transforms SSL from a pretraining mechanism into a discovery tool: it directly tests which temporal resolutions carry predictive signal for downstream tasks. In doing so, HiMAE preserves the efficiency of masked autoencoding while introducing interpretability absent in contrastive or flat masked objectives.

**Multi-scale Learning** The emphasis on resolution awareness connects naturally to multi-scale learning, where modeling temporal signals across multiple granularities has emerged as a powerful inductive bias. In vision, multi-scale architectures such as pyramidal CNNs and hierarchical attention enable models to integrate fine-scale edges with coarse semantic structures, substantially improving recognition and generation in 2D (Wang et al., 2016; Yang et al., 2016; Liu et al., 2021a; Kusupati et al., 2024; Liu et al., 2024) and 3D (He et al., 2017; Ghadai et al., 2019; Zhang et al., 2022b).

In time series, multi-scale methods are fewer but increasingly influential. N-HiTS (Challu et al., 2022) improves long-horizon forecasting by allocating capacity across frequencies via hierarchical interpolation. Pyraformer (Liu et al., 2022) leverages pyramidal attention to capture dependencies over a tree of scales, while Scaleformer (Shabani et al., 2023) introduces iterative refinement across resolutions. Pathformer (Chen et al., 2024) further adapts pathways dynamically to match input-specific temporal dynamics.

Prior multi-scale methods typically rely on fixed hierarchies or task-specific refinement stages (e.g., for forecasting), which constrains their generality. While HiMAE also inherits inductive biases from convolutional design choices (e.g., step size, padding, kernel width), these parameters define receptive fields rather than dictate which scales are salient. By coupling self-supervised reconstruction with these fields, HiMAE induces a hierarchy of temporal embeddings that can be probed independently.

## 3 METHODS

**Hierarchical Masked Autoencoders (HiMAE)**  HiMAE combines masked autoencoding (Baldi, 2012; He et al., 2022) with 1-D physiological time series by coupling a patch-masking objective with a U-Net–style convolutional encoder–decoder (Ronneberger et al., 2015). Given an input sequence $x \in \mathbb{R}^{C \times L}$, we partition it into $N = L/P$ non-overlapping patches of length $P$. A binary mask $m \in 0, 1^N$ is sampled from a Bernoulli distribution with parameter $r$, indicating the masking ratio. Masked indices are selected uniformly at random without replacement, expanded to match temporal resolution as $m' \in 0, 1^L$, and applied to the sequence, yielding $\tilde{x} = x \odot (1 - m')$. This masking procedure removes substantial context, forcing the model to infer higher-order dependencies. In addition to random masking, we also employ contiguous masking, in which adjacent patches are removed to mimic sensor dropout similar to recent protocols showing benefits (Xu et al., 2025). Both regimes are interleaved during pretraining to promote robustness across reconstruction settings.

**Architecture**  The encoder $f_\theta$ is a hierarchical 1D CNN composed of residual convolutional blocks with stride-2 convolutions that downsample the temporal resolution by half at each stage, expanding the receptive field so that deeper layers capture long-range dependencies while shallow layers retain local detail. Each residual block consists of two convolutions with kernel size 5, batch normalization (Ioffe & Szegedy, 2015), and GELU activations (Hendrycks & Gimpel, 2023), along with a projection shortcut when input and output dimensions differ. The decoder $g_\phi$ mirrors this structure with transposed convolutions for upsampling and incorporates skip connections from encoder layers, concatenating intermediate features to restore fine-grained temporal structure. All convolutions are standard 1D operations defined over temporal windows, and striding handles subsampling directly. Intermediate activations use GELU, while the final layer applies a $\tanh$ nonlinearity so that outputs $\hat{x} \in \mathbb{R}^{C \times L}$ are bounded in $[-1, 1]$, matching the normalized input range.

We deliberately adopt a convolutional U-Net backbone rather than a transformer-based encoder for two reasons. First, physiological signals exhibit strong local dependencies governed by morphology (e.g., PPG waveform shape, ECG peaks), which are naturally modeled by finite receptive fields. Convolutions (O'Shea & Nash, 2015) encode this locality directly, whereas transformers must simulate it through restricted attention, often at higher parameter cost. Second, multi-resolution structure is intrinsic to physiology (e.g., heartbeats unfold over milliseconds, rhythms span seconds). A hierarchical CNN with skip connections provides an architectural bias toward such nested timescales, aligning directly with the resolution hypothesis and being orders of magnitude smaller than other proposed foundation models in this space (See Figure 2 for comparison). In contrast, transformers emphasize global mixing, which may obscure resolution-specific structure while consuming substantially more compute (Table 5). This rationale motivates HiMAE's design as not only efficient but also inductively aligned with the temporal statistics of wearable signals.

**Resolution Probes**  Multi-resolution embeddings extracted from different levels of the hierarchy are probed independently, with distinct linear classifiers trained per resolution (Alain & Bengio, 2018). This design enables us to systematically evaluate which temporal granularity carries predictive signal for downstream tasks, rather than collapsing embeddings into a single latent space. Finally, choices of patch length $P$ and kernel size were guided by ablations (Appendix Section F.1), which confirmed that $P = 5$ and kernel size 5 yield the best balance between local fidelity and receptive field expansion when all other hyperparameters were fixed.

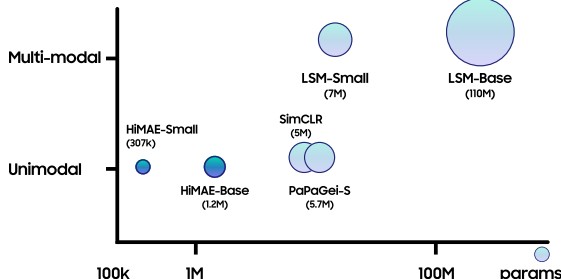

Figure 2: **HiMAE is lightweight**

Training minimizes a masked reconstruction loss restricted to occluded regions: $\mathcal{L}_{\mathrm{MSE}}(\theta, \phi) = \frac{\|(\hat{x}-x) \odot m'\|_2^2}{\sum_{t=1}^L m'_t}$, where $m'$ ensures that gradients are only computed on masked segments. This objective estimates $p(x_\mathcal{M}|x_\mathcal{O})$, with $\mathcal{M}$ and $\mathcal{O}$ denoting masked and observed indices, preventing trivial copying of visible inputs and promoting temporally coherent, multi-scale representations.

**Effective Global Context via Receptive Field Expansion** While Transformers achieve global dependency modeling via $O(L^2)$ self-attention, the U-Net architecture in HiMAE approximates this behavior at $O(L)$ complexity through hierarchical spatial contraction. In a $D$-layer encoder, the effective receptive field (ERF) at layer $d$ grows exponentially as $R_d = R_{d-1} + (k-1) \cdot \prod_{i=1}^{d-1} s_i$, where $k$ is the kernel size and $s$ is the stride. By the bottleneck, the ERF encompasses a significant portion of the input sequence $L$, allowing the model to capture "global" context without the quadratic memory overhead of an attention matrix. This hierarchical aggregation acts as a multi-scale proxy for global attention: deep layers integrate coarse, long-range context, while skip connections inject high-resolution local features back into the decoder. Consequently, HiMAE simulates the communicative benefits of attention through a series of local-to-global inductive biases, achieving competitive representation power at a fraction of the FLOPs required by vanilla ViT or Transformer-based autoencoders.

**Pretraining and Evaluation Protocol** PPG Sequences were sampled at $f_s = 100$ Hz over fixed windows of $T = 10$s ($L = 1000$ timesteps). 10 second windows were selected due to clinically actionable events occurring in these time scales (ECG is collected at 10s intervals in clinical settings (Shuai et al., 2016; Elgendi, 2012)) and due to our interest in real-time monitoring on edge devices. Each signal was divided into non-overlapping patches of length $P = 5$ (200 patches total), and a masking ratio $r = 0.8$ was applied with patterns resampled per sequence and iteration to mitigate overfitting (we empirically tested this masking ratio in Appendix Section F.1 with similar observations made in (Narayanswamy et al., 2024)). The encoder architecture employed channel widths $[16, 32, 64, 128]$, mirrored in the decoder. Optimization was performed with AdamW (Loshchilov & Hutter, 2019) (lr $= 10^{-3}$, weight decay $= 10^{-3}$) using a warmup–cosine schedule (10% linear warmup steps followed by cosine decay). Models trained up to 100k steps with batch size 2048 and early stopping triggered after 3 epochs without improvement similar to the protocols found in (Narayanswamy et al.). Data splits followed a 90/10 (train/validation) protocol across subjects, ensuring no identity overlap between pretraining and validation. Pretraining converged within 12 hours when distributing training across 4 Tesla T4 GPUs using PyTorch lightning (Paszke et al., 2019).

**Pretraining datasets.** We construct our pretraining corpus from approximately $80,000$ hours of wearable green PPG signals, drawn from seven large-scale free world studies conducted at Samsung Research. These datasets include recordings from $47,644$ participants across seven distinct wearable devices, capturing broad demographic, behavioral, and hardware variability in a noisy environment (See Appendix Section B for ethics considerations). Although our modeling framework is modality-agnostic and can extend to other physiological signals such as electrocardiograms (see Appendix F.2), we focus here on PPG due to its prevalence and the scale of available data (we lack the same order of magnitude of ECG compared to PPG because ECG is not passively collected). To ensure reliability, we apply a standardized preprocessing pipeline that retains only high-quality segments, filtering by a Signal Quality Index (SQI). The retained signals are further refined using a bandpass filter of 0.5–8 Hz (Christiano & Fitzgerald, 2003), consistent across all pretraining and evaluation studies, to isolate physiologically relevant dynamics. Finally, signals are normalized to the range $[-1, 1]$ to match the output range of the tanh activation function used in our models.

## 4 Experimental Design

We follow the evaluation protocol of Narayanswamy et al. (2024) and extend it into a unified benchmark suite spanning generative, and classification, along with ablations to quantify how key architectural components interact with scaling. Across all experiments, our goal is not only to assess HiMAE's efficiency and transferability, but also to test the *resolution hypothesis*: whether predictive signal concentrates at specific levels of the hierarchical embeddings. Further analysis and results are displayed in full in Appendix Section F.

**Model scaling and generative reconstruction.** We first study HiMAE's scaling properties by measuring how reconstruction performance varies as a function of dataset size, number of participants, model capacity, and training compute capacity (batch size). For each axis, we systematically subsample or expand the relevant resource while holding others fixed, enabling us to isolate its contribution to representation quality. Scaling is assessed through mean squared error on masked reconstruction on a held out validation set, which provides a direct measure of how model capacity and data availability govern loss reduction. We also squeeze in ablations in this experiment to assess how removing skip connections, and removing the hierarchal design affect scaling.

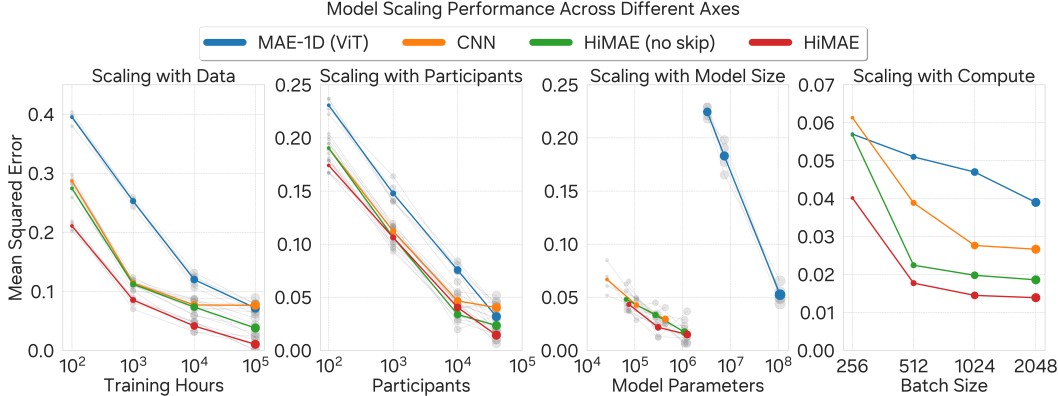

Figure 3: **HiMAE exhibits superior scaling across axes.** Mean squared error decreases most rapidly for HiMAE as data, participants, model size, and compute scale on a held out validation set. Ablations without skip connections confirm that both the hierarchical design and skip pathways are helpful for generative performance. Grey lines indicate multiple runs whereas colored lines are average performance.

To complement this aggregate view, we also evaluate generative performance under three increasingly challenging reconstruction regimes defined in the LSM papers (Narayanswamy et al.; Xu et al., 2025): (i) random imputation, where patches are masked at random uniformly; (ii) temporal interpolation, where contiguous spans are removed to simulate sensor dropout; and (iii) temporal extrapolation, where future spans are occluded and predictions must rely solely on past context. We compute the mean squared error (MSE) for these evaluations.

**Classification** To assess downstream transferability and adaptability, we benchmark HiMAE on 12 binary classification tasks drawn from labeled datasets fully disjoint from our pretraining sources. We organize these into three groups: cardiovascular outcomes, sleep staging, and abnormal laboratory prediction. Cardiovascular outcomes, provide the most established benchmarks, with well-documented links between PPG and clinical endpoints (Shabaan et al., 2020). These include hypertension detection, and arrhythmia-related events such as Premature Ventricular Contractions (PVCs) detections, typically identified via electrocardiograms (ECGs). Sleep staging is another task we include which is of high interest, given the demand for wearables to track fine-grained sleep states despite the temporal and physiological complexity of the task (Imtiaz, 2021; Thapa et al., 2024; Birrer et al., 2024). Laboratory predictions, on the other hand, serves as a discovery setting, testing whether PPG contains sufficient biomarker information to separate abnormal from healthy labs—an open question compared to patient-record benchmarks where such signals are more explicit (Kolo et al., 2024; McDermott et al., 2025). Together, these canonical and exploratory tasks form a spectrum that enables a comprehensive evaluation of representation quality across diverse digital health applications. All tasks are described in greater detail in Appendix Section D.

We evaluate HiMAE against two complementary classes of baselines. The first comprises established self-supervised representation learning methods for time series, including SimCLR (Chen et al., 2020b), DINO (Caron et al., 2021), Masked Siamese Networks (MSN) (Assran et al., 2022), and a ViT-based 1D masked autoencoder that follows the LSM training protocol of (Narayanswamy et al.). The second class consists of state-of-the-art time-series and wearable foundation models. This includes PaPaGei, a leading foundation model for PPG (Pillai et al., 2024), evaluated both using its publicly released Bell Labs checkpoint (PaPaGei-BL)[1] and a variant retrained on our pretraining corpus (PaPaGei-SRA). We additionally benchmark against Chronos (Ansari et al., 2024), a large-scale time-series foundation model, and the hierarchical Swin Transformer (Liu et al., 2021b), configured to match the LSM setting for controlled comparison. Further implementation details for all baselines are provided in Appendix E. All models are evaluated using a standard linear probing protocol, in which the pretrained encoder is frozen and a linear classifier is trained on top of the learned representations. Performance is reported using AUROC as the primary metric of discriminative ability. For every architecture, we expose the full sequence of embeddings along the temporal

---

[1]https://zenodo.org/records/13983110

Table 1: **Performance on generative benchmarks.** Mean squared error and $r^2$ for random imputation, temporal interpolation, and temporal extrapolation at varying missingness levels. Bold outline indicates best performing model.

| (a) Random Imputation (MSE) | | | | (b) Temporal Interpolation (MSE) | | | | (c) Temporal Extrapolation (MSE) | | |
|---|---|---|---|---|---|---|---|---|---|---|
| Method | 30% | 50% | 80% | Method | 30% | 50% | 80% | Method | 30% | 50% | 80% |
| Mean Fill | 0.224 | 0.225 | 0.225 | Mean Fill | 0.223 | 0.224 | 0.225 | Mean Fill | 0.225 | 0.225 | 0.225 |
| NN Fill | 0.023 | 0.137 | 0.188 | NN Fill | 0.451 | 0.466 | 0.491 | NN Fill | 0.450 | 0.493 | 0.529 |
| Linear Int. | 0.045 | 0.083 | 0.153 | Linear Int. | 0.377 | 0.385 | 0.403 | Linear Int. | 0.449 | 0.487 | 0.526 |
| MAE-1D (ViT) | 0.017 | 0.029 | 0.041 | MAE-1D (ViT) | 0.173 | 0.203 | 0.299 | MAE-1D (ViT) | 0.255 | 0.282 | 0.356 |
| CNN | 0.017 | 0.030 | 0.040 | CNN | 0.165 | 0.199 | 0.278 | CNN | 0.248 | 0.269 | 0.343 |
| HiMAE | **0.012** | **0.017** | **0.026** | HiMAE | **0.126** | **0.173** | **0.201** | HiMAE | **0.194** | **0.202** | **0.211** |

| (d) Random Imputation (R$^2$) | | | | (e) Temporal Interpolation (R$^2$) | | | | (f) Temporal Extrapolation (R$^2$) | | |
|---|---|---|---|---|---|---|---|---|---|---|
| Method | 30% | 50% | 80% | Method | 30% | 50% | 80% | Method | 30% | 50% | 80% |
| Mean Fill | — | — | — | Mean Fill | — | — | — | Mean Fill | — | — | — |
| NN Fill | 0.897 | 0.391 | 0.164 | NN Fill | -1.022 | -1.080 | -1.182 | NN Fill | -1.000 | -1.191 | -1.351 |
| Linear Int. | 0.799 | 0.631 | 0.320 | Linear Int. | -0.691 | -0.719 | -0.791 | Linear Int. | -0.996 | -1.164 | -1.338 |
| MAE-1D (ViT) | 0.924 | 0.871 | 0.818 | MAE-1D (ViT) | 0.224 | 0.094 | -0.329 | MAE-1D (ViT) | -0.133 | -0.253 | -0.582 |
| CNN | 0.924 | 0.867 | 0.822 | CNN | 0.260 | 0.112 | -0.236 | CNN | -0.102 | -0.196 | -0.524 |
| HiMAE | **0.946** | **0.924** | **0.884** | HiMAE | **0.435** | **0.228** | **0.107** | HiMAE | **0.138** | **0.102** | **0.062** |

dimension, rather than collapsing representations to a single summary token, ensuring that downstream probes retain access to resolution-specific information. This evaluation protocol allows us to assess whether pretraining yields representations that are both discriminative and transferable across tasks.

**Resolution Hypothesis** HiMAE produces embeddings at multiple temporal scales, and we probe each scale independently with linear classifiers. This allows us to test whether predictive information is concentrated at fine, intermediate, or coarse resolutions depending on the clinical endpoint. In this way, the classification tasks serve not only as benchmarks for transfer learning, but also as controlled tests of the resolution hypothesis (Receptive field lengths are described in Section C.1).

## 5 RESULTS

### 5.1 SCALING AND GENERATIVE BENCHMARK

**Scaling:** We first examine the scaling behavior in Figure 3 of HiMAE relative to baselines across data, participants, model parameters, and compute capacity (batch size). The overall scaling trends follow conventional expectations, error decreases monotonically with additional data, participants, or compute. However, scaling with model parameters reveals a interesting insight. HiMAE achieves substantially lower loss at smaller parameter capacities, while transformers only begin to close the gap once scaled to orders of magnitude more parameters (we chose transformer parameter count based on LSM's original paper (Narayanswamy et al., 2024)). This difference reflects an inductive bias. Transformer which assume global receptive fields, appear to require considerably larger model capacity before capturing the local dynamics of the data. In contrast, HiMAE's hierarchical convolutional structure exploits spatial and temporal locality efficiently, yielding superior performance at modest scales. This observation reinforces the importance of architectural priors in low-capacity regimes.

**Generative:** Turning to generative benchmarks, HiMAE consistently outperforms all baselines across random imputation, temporal interpolation, and temporal extrapolation tasks (Table 1). In terms of mean squared error, HiMAE achieves the lowest reconstruction error in every setting, including cases with heavy missingness. This advantage persists when evaluated with $R^2$, where the mean-fill baseline serves as the reference. By achieving positive $R^2$ scores even in challenging extrapolation scenarios, HiMAE demonstrates reconstruction ability beyond naive heuristics (e.g., mean fill, nearest neighbor, or linear interpolation). Together, these results establish HiMAE as a strong generative model for missing data problems, with advantages that persist across scaling regimes and input corruption patterns.

**Ablations:** Ablation in Table 1 and Figure 3 further highlights the contributions of hierarchical design and skip connections in HiMAE. Removing either component results in increased error, indicating that both are crucial for effective representation learning. Nevertheless, even without these architectural elements, HiMAE variants remain competitive with larger transformer based models,

Table 2: Linear probing classification performance comparison against baselines on different tasks. AUROC is reported in percent with 95% confidence intervals. The best performance is **bold**, the second best model is underscored. * denotes $p < 0.05$, ** denotes $p < 0.01$ from a two-sided z-test comparing HiMAE with the second-best model.

| Model | #param (M) | Cardiovascular Conditions | | | Abnormal Blood Labs | | | | | Sleep Staging | | | |
|---|---|---|---|---|---|---|---|---|---|---|---|---|---|
| | | Hyptn (lab) | Hyptn (free-living) | PVC | A1C | Hemoglobin | Platelets | Sodium | Potassium | Wake | Light | Deep | REM |
| SimCLR | 5.0 | 53.4 (±3.6) | 53.7 (±4.1) | 51.7 (±5.9) | 60.9 (±5.5) | 50.8 (±4.6) | 44.4 (±6.2) | 58.6 (±4.8) | 67.0 (±4.9) | 56.6 (±4.3) | 52.7 (±5.1) | 67.0 (±4.0) | 51.0 (±5.7) |
| DINO | 6.5 | 51.7 (±4.8) | 52.2 (±3.4) | 47.0 (±4.8) | 58.9 (±3.9) | 49.6 (±3.2) | 42.9 (±5.4) | 56.9 (±3.3) | 64.5 (±3.8) | 55.3 (±3.6) | 55.2 (±4.4) | 68.8 (±3.3) | 46.0 (±6.0) |
| MSN | 2.5 | 55.2 (±2.8) | 55.2 (±2.5) | 56.4 (±3.0) | 62.9 (±2.2) | 52.1 (±2.4) | 45.9 (±3.7) | 60.4 (±2.1) | 69.5 (±2.3) | 57.8 (±2.7) | 50.3 (±2.9) | 65.3 (±2.8) | 56.0 (±3.5) |
| MAE (ViT) | 110.6 | 43.2 (±7.0) | 65.0 (±5.5) | 72.2 (±7.0) | **79.6** (±6.5) | **57.6** (±4.9) | 56.1 (±5.8) | 48.8 (±6.1) | 76.5 (±6.8) | 63.8 (±5.3) | **60.8** (±5.8) | 69.3 (±6.6) | **59.7** (±6.2) |
| HiMAE | 1.2 | **65.1**\*\* (±1.7) | 65.1 (±1.6) | **80.2**\* (±1.4) | 70.1 (±2.0) | 56.2 (±1.3) | **68.5**\*\* (±1.8) | **63.3**\* (±1.9) | **83.1** (±1.5) | **66.8** (±1.8) | 59.3 (±2.1) | **72.3** (±1.4) | 58.5 (±2.2) |

Table 3: Linear probing classification performance comparison against state-of-the-art wearable and time-series foundation models. AUROC is reported in percent with 95% confidence intervals. The best performance is **bold**, the second best model is underscored. * denotes $p < 0.05$, ** denotes $p < 0.01$ from a two-sided z-test comparing HiMAE with the second-best model.

| Model | #param (M) | Cardiovascular Conditions | | | Abnormal Blood Labs | | | | | Sleep Staging | | | |
|---|---|---|---|---|---|---|---|---|---|---|---|---|---|
| | | Hyptn (lab) | Hyptn (free-living) | PVC | A1C | Hemoglobin | Platelets | Sodium | Potassium | Wake | Light | Deep | REM |
| PaPaGei-BL | 5.7 | 57.3 (±4.7) | 60.9 (±4.1) | 74.2 (±6.4) | 59.2 (±5.8) | 58.5 (±5.2) | 59.9 (±4.9) | 59.0 (±4.3) | 75.5 (±5.5) | 56.8 (±4.9) | 55.6 (±5.0) | 53.9 (±4.5) | 56.3 (±5.7) |
| PaPaGei-SRA | 5.7 | 59.3 (±3.5) | 62.9 (±3.7) | 75.2 (±5.6) | 61.2 (±4.1) | **60.5** (±2.7) | 61.9 (±3.6) | 61.0 (±3.3) | 77.5 (±4.6) | 56.8 (±4.2) | 57.6 (±3.7) | 55.9 (±3.4) | 58.3 (±5.1) |
| Swin-Transformer | 110.6 | 58.3 (±6.2) | 61.9 (±5.8) | 74.2 (±7.2) | 58.2 (±6.7) | 59.5 (±8.0) | 60.9 (±7.1) | 60.0 (±5.6) | 76.5 (±6.8) | 56.7 (±5.4) | 54.7 (±6.1) | 53.3 (±7.5) | 54.4 (±5.3) |
| Chronos | 200.0 | **67.3** (±1.7) | 59.9 (±2.9) | 65.7 (±3.1) | 58.2 (±3.4) | 53.6 (±3.3) | 60.9 (±2.7) | **63.3** (±2.3) | 63.5 (±2.8) | 64.9 (±2.7) | **63.2** (±2.1) | 72.2 (±2.5) | 57.3 (±2.9) |
| HiMAE | 1.2 | 65.1 (±1.7) | **65.3** (±1.6) | **80.2** (±1.4) | **70.1**\*\* (±2.0) | 56.2 (±1.3) | **68.5**\*\* (±1.8) | 63.3 (±1.9) | **83.1**\* (±1.5) | **66.8** (±1.8) | 59.3 (±2.1) | **72.3** (±1.4) | 58.5 (±2.2) |

underscoring the robustness of the approach. More importantly, the full model exhibits improved generalization across scaling axes (Appendix Section F.3), suggesting that the combination of hierarchy and skip connections facilitates better transfer as data and compute grow.

## 5.2 CLASSIFICATION BENCHMARKING

**Classification** In Tables 2 and 3, HiMAE consistently secures the majority of wins, frequently outperforming or matching models that are considerably larger. This is particularly striking given that prior work has typically relied on heavy architectures to reach similar levels of performance, highlighting HiMAE's ability to capture a broad spectrum of physiological features with a compact design. These outcomes emphasize the model's robustness when applied to structured, temporally dependent problems that demand sensitivity to subtle variations in wearable signals.

Taken together, these results position HiMAE as the most consistently strong performer across the benchmark suite. In cases where HiMAE does not place first it is only ∼1-2% behind the winning model. Crucially, this level of performance is achieved with a substantially smaller model than competing approaches, demonstrating a favorable tradeoff between efficiency and predictive power. Rather than excelling only in isolated cases, HiMAE delivers broad, cross-domain competitiveness, suggesting that compact models, when designed with the right inductive biases, can rival or even surpass far larger architectures.

## 5.3 RESOLUTION SPECIFIC CLINICAL INTERPRETABILITY

The resolution hypothesis predicts that different health outcomes depend on distinct temporal granularities. To test this, we analyze performance across HiMAE layers, where each layer corresponds to a progressively coarser resolution. Figure 4 reveals clear resolution-specific structure: individual downstream tasks achieve maximal AUROC at different layers, highlighted by the red boundaries.

This layer-task alignment underscores two key insights. First, temporal resolution is not a nuisance parameter but an axis of predictive structure: different outcomes are best represented at different scales (we show that collapsing an encoder decoder still has concordant results showing that our hierarchal model is not an artifact in Appendix Section F.4). Second, HiMAE naturally exposes this heterogeneity, functioning as a discovery tool for identifying the most informative resolution per task. This complements conventional interpretability methods (Amann et al., 2022; Xu et al., 2023; Lee et al., 2025) by shifting the focus from *which features* drive predictions to *which resolutions* matter. In doing so, HiMAE operationalizes the resolution hypothesis and provides insights to tasks where the resolution needed is not entirely clear.

**Clinical Interpretation**

The resolution-specific structure discovered by HiMAE carries clinical implications that resonate with existing literature or provide new clinical insights from a scientific discovery perspective. Car-

**HiMAE Layers Discover Resolution-Specific Structure Across Downstream Tasks**

| | Cardiovascular | | | Blood-related labs | | | Electrolytes | | Sleep Staging | | | Corresponding Receptive Field Resolutions |
|---|---|---|---|---|---|---|---|---|---|---|---|---|
| HiMAE-L1 | 0.58 | 0.62 | 0.73 | 0.70 | 0.52 | 0.68 | 0.55 | 0.70 | 0.63 | 0.56 | 0.66 | 0.56 |
| HiMAE-L2 | 0.57 | 0.62 | 0.72 | 0.66 | 0.56 | 0.65 | 0.59 | 0.72 | 0.64 | 0.58 | 0.69 | 0.57 |
| HiMAE-L3 | 0.60 | 0.64 | 0.77 | 0.65 | 0.53 | 0.65 | 0.59 | 0.82 | 0.66 | 0.59 | 0.72 | 0.58 |
| HiMAE-L4 | 0.65 | 0.65 | 0.80 | 0.68 | 0.52 | 0.66 | 0.63 | 0.83 | 0.66 | 0.59 | 0.71 | 0.58 |
| HiMAE-L5 | 0.62 | 0.65 | 0.80 | 0.70 | 0.52 | 0.66 | 0.59 | 0.82 | 0.66 | 0.59 | 0.70 | 0.58 |

*HiMAE Model & Layer* (y-axis)

Downstream Tasks (x-axis): Hypertension (Lab), Hypertension (Free), PVC Detection, A1C, Hemoglobin, Platelets, Sodium, Potassium, Wake, Light, Deep, REM

Figure 4: **HiMAE discovers task-specific structures for downstream tasks.** AUROC across layers shows that tasks rely on distinct temporal scales, highlighting HiMAE as a tool for discovering the most informative resolution in clinical machine learning.

diovascular and sleep-staging outcomes achieve maximal performance in later layers, which aligns with physiological understanding: cardiovascular (Zhou et al., 2021; Tang et al., 2025) and sleep dynamics (Patanaik et al., 2018), which evolve gradually over longer time horizons. Capturing these slower patterns requires less temporal granularity, consistent with the notion that general trends, not transient spikes, dominate predictive structure in chronic or cyclic physiological processes.

In contrast, tasks involving laboratory measurements, are a more exploratory and scientific discovery setting where there isn't much intuition on what resolution should reveal the most predictive signal. When looking at Figure 4, particularly the blood-related biomarkers, it exhibit optimal performance in earlier layers corresponding to finer temporal scales. These outcomes reflect inherently volatile physiological processes, where shifts in morphology can signal meaningful physiological change on lab measurements.

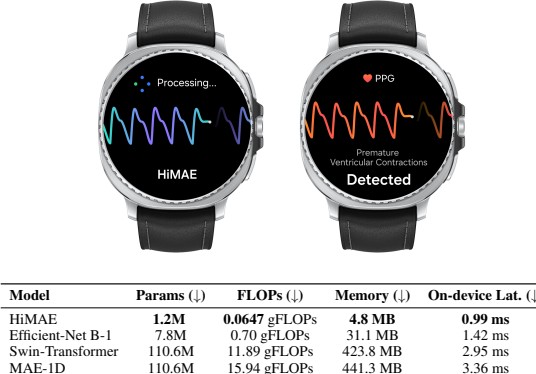

| Model | Params (↓) | FLOPs (↓) | Memory (↓) | On-device Lat. (↓) |
|---|---|---|---|---|
| HiMAE | **1.2M** | **0.0647** gFLOPs | **4.8 MB** | **0.99 ms** |
| Efficient-Net B-1 | 7.8M | 0.70 gFLOPs | 31.1 MB | 1.42 ms |
| Swin-Transformer | 110.6M | 11.89 gFLOPs | 423.8 MB | 2.95 ms |
| MAE-1D | 110.6M | 15.94 gFLOPs | 441.3 MB | 3.36 ms |

Figure 5: **Model efficiency and on-device inference:** Sample on-device detections on Samsung Galaxy device. Size, compute cost, memory footprint, and CPU latency (ms per sample, batch size 2048) measured over a 10s sequence at 100Hz.

## 5.4 CASE STUDIES

**Case Study 1: On-Device Benchmarking** A central novelty of HiMAE is that it is, to our knowledge, the first SSL method compact enough to run entirely *on-watch*, rather than on phone-class hardware. We evaluate on-device PVC detection on smartwatch-class CPUs sampled at 100 Hz (Figure 5). HiMAE is exceptionally lightweight (1.2M parameters, 0.0647 gFLOPs, 4.8 MB) and achieves 0.99 ms latency per sample, equivalent to processing ≈1,010 samples/s or ≈2.8 hours of signal per minute of wall time. By contrast it shows massive performance gains against transformer baselines, Swin-Transformer (110M parameters, 11.9 gFLOPs, 423 MB) and a MAE-1D (ViT) (110M, 15.9 gFLOPs, 441 MB). HiMAE also outperforms optimized models like Efficient-Net B1 (Tan & Le, 2020) providing context to the latency and compactness of our model. HiMAE is thus ∼3–4× more efficient compared to transformers while fitting fully on-watch (without quantization (Jacob et al., 2017)), enabling continuous, private inference at the point of signal collection. *This prototype is strictly for research and is not deployed commercially.*

**Case Study 2: HiMAE is adaptable in few shot settings**

A central challenge in the wearable domain is that labels are scarce across tasks. Models that can adapt quickly from generic pretraining to specific detection tasks with limited supervision are therefore essential. Figure 6 illustrates this setting: HiMAE provides strong representations that can be adapted to diverse tasks such as PVC detection or hypertension monitoring with only a handful of labeled examples as reflected by the shape of the learning curves on the few-shot learning experiments. By reducing the supervision required to reach high performance, HiMAE enables new tasks to be supported on-device without the prohibitive cost of large curated datasets which help bolster its practical utility.

Figure 6: **Few-shot adaptation.** HiMAE adapts efficiently to new wearable tasks under sparse labels indicated by curve shape over transformer baselines.

## 6 DISCUSSION

**Summary.** HiMAE advances wearable self supervised methods along three dimensions: (i) its flexible architecture is expressly designed for multi-resolution mapping, enabling seamless adaptation across heterogeneous tasks, (ii) by aligning task-dependent resolutions with model representations, it not only optimizes predictive performance but also offers a window into the temporal organization of physiological biomarkers, and (iii) by design of the compactness, it achieves the first demonstration of true *on-watch* inference, running entirely within smartwatch-class constraints while matching or surpassing performance on far larger models. These results position HiMAE as an efficient representation learner but also as a framework for interrogating which temporal resolutions carry signal.

**Resolution as a structural prior.** Our findings validate the resolution hypothesis and suggest a shift in how representation learning on wearables should be conceptualized. This reframing implies that representation learning for physiological signals should expose, rather than collapse, scale-specific embeddings. The layer-wise AUROC profiles in Figure 4 show that predictive performance peaks at different levels of the hierarchy depending on the task, with fine-scale embeddings capturing short-lived physiological events and coarse-scale embeddings capturing slower behavioral phenomena. By revealing this heterogeneity, HiMAE provides empirical evidence that resolution-specific representations are essential for wearable health modeling.

**From "on-device" to "on-watch."** HiMAE demonstrates that convolutional hierarchies can reduce model size by two orders of magnitude relative to transformer-based models, enabling the first instance of true *on-watch* inference. This moves the deployment frontier from phone-class to watch-class processors, where inference occurs exactly at the point of sensing. Beyond efficiency, this shift has consequences for privacy (data never leave the device) and for clinical viability (continuous real-time monitoring becomes feasible).

**Limitations and Future Works** While we focus on PPG, the principles underlying HiMAE generalize to multimodal settings. Physiological signals are inherently multi-scale across modalities (e.g., ECG beats, accelerometer motion cycles, EEG rhythms), and resolution-aware architectures could expose complementary temporal signatures across them. Another limitation of our work is we don't handle sequences beyond 10 second windows which could unlock another breadth of tasks. Future works also warrants a clinical validation to the discoveries made by HiMAE which could be of significant interest to the health community.

## LLM USAGE

A large language model (LLM) was used to assist in refining the phrasing and structure of the manuscript. Its role was limited to improving clarity, coherence, and readability of the text based on author-provided drafts. All scientific content, experimental design, and analysis were conceived, implemented, and verified by the authors.

## ACKNOWLEDGMENTS

We thank Minji Han and Rachel Choi for their expertise in UX/UI design and for crafting the specialized visualizations not supported by standard Python libraries; their design contributions were essential to this work. We also thank Praveen Raja, Matthew Wiggins, Yilin Shen, and Mike Freedman for their invaluable feedback and insightful discussions throughout the project.

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

APPENDIX

## A    AUTHOR CONTRIBUTION

We attribute proper credit to the following authors for the development of this project

Table 4: Overview of author contributions.

| Author | Concept | Experiment Design | Coding | Analysis | Writing | Visualization | Project Mgmt. | Discussion | Resources |
|---|---|---|---|---|---|---|---|---|---|
| Simon Lee | ✓ | ✓ | ✓ | ✓ | ✓ | ✓ | ✓ | ✓ | |
| Cyrus Tanade | | | ✓ | ✓ | ✓ | | ✓ | ✓ | ✓ |
| Hao Zhou | | | ✓ | ✓ | | ✓ | | ✓ | |
| Juhyeon Lee | | | | ✓ | ✓ | | | ✓ | ✓ |
| Megha Thurkal | | | | ✓ | | | | ✓ | ✓ |
| Minji Han | | | | | | ✓ | | ✓ | ✓ |
| Baiying Liu | | | | ✓ | | | | ✓ | |
| Keum San Chun | | | | | | | | ✓ | ✓ |
| Migyeok Gwak | | | | | | | | ✓ | ✓ |
| Mehrab Bin Morshed | | | | | | | | ✓ | |
| Viswam Nathan | | | | | | | | ✓ | |
| Mahbubur Rahman | | | | | | | | ✓ | ✓ |
| Li Zhu | | | | | | | | ✓ | |
| Sharanya Desai | | ✓ | | | ✓ | | ✓ | ✓ | ✓ |

## B    ETHICS CONSIDERATIONS

### B.1    DATA PRIVACY AND CONSENT

Wearable signals capture sensitive physiological and behavioral information (Erturk et al., 2025). Our study relies on both publicly available and proprietary (company-owned) datasets that have been carefully vetted. These datasets include transparent disclosure of data usage, explicit opt-in mechanisms, and the option for participants to withdraw (Perez-Pozuelo et al., 2021). Across the seven datasets used in this study, we obtained written consent (via paper or digital waivers) that clearly informed participants that their data may be used for commercial research purposes.

### B.2    BIAS AND REPRESENTATIVENESS

Physiological signals vary across age, gender, ethnicity, health status, and socioeconomic context, yet most existing datasets underrepresent key populations (FitzGerald & Hurst, 2017; McCradden et al., 2020; Chen et al., 2021). Such underrepresentation risks embedding biases into foundation models, leading to inequitable performance in downstream applications. Mitigation requires deliberate corpus curation, bias auditing, and systematic evaluation across diverse cohorts. In this study, we sought to mitigate bias by incorporating a pre-training corpus drawn from a wide range of wearable devices, collected across multiple regions of the world and over many years.

### B.3    CLINICAL IMPLICATIONS

Wearable foundation models are not substitutes for medical judgment. Their predictions require regulatory approval and clinical validation before integration into healthcare practice. Without safeguards, model misinterpretation could lead to misdiagnosis or inappropriate treatment. Development should involve clinical collaborators, real-world evaluations, and explicit positioning of models as decision-support rather than diagnostic systems. In our group, ongoing collaborations aim to evaluate where our foundation model performs well and how it may assist in forming clinical insights. We emphasize that no definitive clinical conclusions should be drawn from this work.

### B.4    ENVIRONMENTAL IMPACT

Training generative models entails substantial computational and environmental costs (Ligozat et al., 2022; Bender et al., 2021; Bouza et al., 2023). To minimize our footprint, we limited redundant runs, and reused checkpoints to avoid unnecessary GPU usage. All experiments were conducted on datacenter GPUs with efficient cooling systems and renewable energy credits to reduce carbon intensity. We emphasize that transparent reporting of compute usage and bounding resource allocation are necessary steps toward sustainable machine learning research.

## C  REPRODUCIBILITY STATEMENT

Table 5: HiMAE architecture components.

Encoder–Decoder

| Layer | Output Shape |
|---|---|
| Input | $[B, 1, T]$ |
| EncoderConvBlock(1→16) | $[B, 16, T/2]$ |
| EncoderConvBlock(16→32) | $[B, 32, T/4]$ |
| EncoderConvBlock(32→64) | $[B, 64, T/8]$ |
| EncoderConvBlock(64→128) | $[B, 128, T/16]$ |
| EncoderConvBlock(128→256) | $[B, 256, T/32]$ |
| DecoderSkipBlock(256→128) | $[B, 128, T/16]$ |
| DecoderSkipBlock(128→64) | $[B, 64, T/8]$ |
| DecoderSkipBlock(64→32) | $[B, 32, T/4]$ |
| DecoderSkipBlock(32→16) | $[B, 16, T/2]$ |
| Final Deconv (16→1) | $[B, 1, T]$ |
| Tanh | $[B, 1, T]$ |

EncoderConvBlock

| Layer |
|---|
| Conv1d ($k = 5$, s=2, p=2) |
| BatchNorm |
| GELU |
| Conv1d ($k = 5$, s=1, p=2) |
| BatchNorm |
| Conv1d ($k = 1$, s=2) + BN |
| GELU |

DecoderSkipBlock

| Layer |
|---|
| ConvTranspose1d ($k = 5$, s=2, p=2, op=1) |
| Concat skip connection |
| Conv1d ($k = 5$, s=1, p=2) |
| BatchNorm |
| GELU |
| Conv1d ($k = 5$, s=1, p=2) |
| BatchNorm |
| GELU |

Due to restrictions around data licensing and industry policies, we are unable to release the full source code associated with HiMAE. However, To mitigate this limitation, we provide a simplified code base in this https://github.com/Simonlee711/HiMAE as well as complete details of the model architecture, layer configurations, and hyperparameters in Table 5. This includes all encoder, decoder, and skip connection blocks, along with kernel sizes, strides, padding, activation functions, and normalization layers. Together, these descriptions and codebases are sufficient to re-implement the model faithfully in any modern deep learning framework (Paszke et al., 2019; Abadi et al., 2016; Bradbury et al., 2018; Hannun et al., 2023). In addition, we report all training settings (e.g., optimizer, learning rate schedule, and batch size) in the Appendix Section E to further support reproducibility. Our goal is to ensure that, while the exact implementation cannot be shared, independent researchers can replicate the methodology and validate the findings presented in this work.

### C.1  TEMPORAL RESOLUTION AS AN EXPLICIT SCALE AXIS

HiMAE's encoder implements a structured mapping from depth to temporal scale. Each encoder block halves the temporal resolution while increasing the span of input samples contributing to each feature, yielding a hierarchy of representations indexed by effective temporal support. This makes temporal resolution an explicit axis of representation, rather than an emergent byproduct of depth.

Concretely, the encoder is composed of $b$ convolutional blocks, each reducing the sequence length by a factor of two. As a result, the representation at depth $b$ operates on a grid of resolution $T/2^b$. At the same time, each block aggregates information over an increasingly large window of the input signal. Because kernel size is fixed across layers, the cumulative temporal support of encoder features grows exponentially with depth, scaling as

$$R_b = \Theta(2^b),$$

up to architecture-dependent constants. Thus, encoder depth simultaneously controls both the granularity at which the signal is represented and the temporal extent over which features are computed.

Table 6 instantiates this scale hierarchy for the HiMAE encoder. Shallow layers operate at high temporal resolution with receptive fields spanning only a few tens of samples, capturing fine-scale waveform morphology. Intermediate layers aggregate information over $10^1$–$10^2$ samples, corresponding to sub-second temporal structure such as beat-to-beat variability. The deepest layers integrate over several hundred samples, encoding longer-range physiological dynamics across multiple cardiac cycles.

This explicit scale stratification is central to masked autoencoding on physiological signals. Because masking removes contiguous temporal regions, successful reconstruction requires contextual information at a scale comparable to the masked interval. Features whose receptive fields are too small lack sufficient context, while features whose receptive fields are too large oversmooth across distinct physiological events. HiMAE's exponential scale ladder ensures that intermediate encoder depths naturally align with the characteristic temporal extent of masked regions, concentrating learning signal at those resolutions.

Table 6: **Temporal resolution and cumulative receptive field through the encoder**. $T$ denotes the input length in samples. $R_\ell$ is the receptive field after layer $\ell$.

| Layer | Kernel $k$ | Stride $s$ | Output length | $R_\ell$ |
|---|---|---|---|---|
| Enc1-conv1 | 5 | 2 | $T/2$ | 5 |
| Enc1-conv2 | 5 | 1 | $T/2$ | 13 |
| Enc2-conv1 | 5 | 2 | $T/4$ | 21 |
| Enc2-conv2 | 5 | 1 | $T/4$ | 37 |
| Enc3-conv1 | 5 | 2 | $T/8$ | 53 |
| Enc3-conv2 | 5 | 1 | $T/8$ | 85 |
| Enc4-conv1 | 5 | 2 | $T/16$ | 117 |
| Enc4-conv2 | 5 | 1 | $T/16$ | 181 |
| Enc5-conv1 | 5 | 2 | $T/32$ | 245 |
| Enc5-conv2 | 5 | 1 | $T/32$ | 373 |

Viewed this way, encoder depth in HiMAE should not be interpreted as a measure of abstraction alone, but as an index over temporal scales. This perspective explains why linear probes trained on intermediate layers often outperform both shallower and deeper representations: they correspond to resolutions at which physiological structure is most predictive.

# D DATASETS

## D.1 AQUISITION AND APPROVAL

All data analyzed in this study were collected under informed consent, with participants explicitly agreeing for their wearable-derived signals to be used in health-related research. The consent language stated that data could be used for developing new health features and algorithms and for inclusion in scientific publications. In particular, participants were informed that health and wellness data such as steps, heart rate, sleep, and photoplethysmography (PPG) signals could contribute to findings aimed at advancing general knowledge of health and science. No data used in this study included personally identifying information such as names or email addresses. We attach a portion of the protocols defined in our user data agreements below:

*The use of these de-identified data for data usage was reviewed and classified as exempt. In addition, because the supporting records constitute case histories and document exposure to devices, we complied with the recordkeeping requirements in 21 CFR § 812.140(a)(3), including obtaining written digital consent and dated information. Participants could withdraw at any time; such withdrawals were documented in the case history, and data collected up to the point of withdrawal were retained and used for the investigation in accordance with the consent and applicable regulations.*

For downstream evaluations, we relied on a combination of institutional review board (IRB)-approved datasets and publicly available resources. For instance, the PVC detection task used paired PPG and ECG recordings to derive annotations of premature ventricular contractions, with ECG-based labels verified both algorithmically and manually. The hypertension classification tasks were drawn from the My Heart Lab Study collected in a lab Setting (ID NCT04314947) and My BP Lab (Clinical Trials ID 19-27169) studies collected in a free-world settting, both of which collected wrist-based PPG alongside reference blood pressure measurements under IRB-approved protocols. Sleep staging was evaluated using the DREAMT dataset, which combines PPG with gold-standard polysomnography annotations in individuals with and without diagnosed sleep disorders. Finally, a range of abnormal lab test prediction tasks were derived from the Tulane University dataset (ID 20242033), linking PPG from Samsung devices with clinical laboratory values for biomarkers (More details in Appendix Section D).

Across all studies, participants consented to data collection through mobile platforms that supported eligibility screening and enrollment, provided full informed consent, and enabled seamless integration of Samsung devices for continuous signal acquisition. Where appropriate, participants also reported medical histories or completed questionnaires through these platforms. All data were de-identified and stored in accordance with the approved study protocols, ensuring compliance with ethical and regulatory standards.

This layered consent and governance framework ensures that the data underpinning our pretraining and evaluation tasks are both ethically sourced and scientifically robust, supporting the broader goal of advancing health monitoring through consumer wearables.

## D.2 PRE-TRAINING DATASETS

Table 7: **Demographic Characteristics of the Study Population.** Distributions are shown by biological sex, age group, racial identity, and BMI category ($N = 47, 644$).

| Category | Subgroup | N | % of total |
|---|---|---|---|
| **Sex** | Male | 36,990 | 77.6 |
| | Female | 10,532 | 22.1 |
| | Another gender identity | 122 | 0.3 |
| **Age** | 18–29 | 12,019 | 25.2 |
| | 30–49 | 27,207 | 57.2 |
| | 50–64 | 7,067 | 14.8 |
| | 65+ | 1,351 | 2.8 |
| **Race** | White | 31,029 | 65.2 |
| | Asian or Pacific Islander | 7,630 | 16.0 |
| | Black or African American | 3,414 | 7.2 |
| | American Indian or Alaskan Native | 592 | 1.2 |
| | Another race | 4,979 | 10.4 |
| **BMI** | Underweight ($< 18.5$) | 823 | 1.7 |
| | Normal weight (18.5–24.9) | 13,626 | 28.5 |
| | Overweight (25–29.9) | 16,634 | 34.9 |
| | Obese I (30–34.9) | 8,745 | 18.5 |
| | Obese II and III ($\geq 35$) | 7,816 | 16.4 |

### D.2.1 DEVICE DISTRIBUTION

The distribution of participants and data availability highlights both the diversity of collection devices and the heterogeneity of study contributions (Figure 7). At the device level, participation is primarily sourced from Watch Active 2, Watch 3, Watch Active, each contributing a lot of participants, while older models such as the Galaxy Gear S3 are represented by fewer users. This heterogeneity in devices provide us with a realistic and diverse set of raw wearable signals that can help us build generalizable foundation models. The presence of entries labeled as "NA" further reflects the mixture of collection devices and the occasional incompleteness of metadata. *We note that the devices used in our study are provided by two distributors limiting its generalizability and causing potential biases due to not having access to other consumer wearable devices.*

### D.2.2 PARTICIPANT COUNTS

In terms of study based segmentation, the dataset contains a handful of large-scale cohort studies, leading to diverse representation (Figure 7). Efforts were made to ensure representation across studies of varying sizes. This underscores the necessity of leveraging the vast scale of high-volume cohorts while simultaneously preserving the heterogeneity introduced by smaller studies, since both dimensions are essential for building foundation models that truly capture the variability and complexity of one-dimensional PPG signal modeling. Our data was collected across 4 countries (USA, South Korea, Brazil, Bangladesh) and the demographics are highlighted in Table 7. Note that missing demographics were imputed via KNN based on average PPG segments which have shown to recover these people specific attributes (Miller et al., 2025; MacIsaac et al., 2025; Ferdinando et al., 2019).

### D.2.3 PRE-PROCESSING PIPELINE

We segment raw PPG signals into fixed-length 10 s windows and apply a lightweight quality-control pipeline to remove motion artifacts and non-physiologic segments. Each window is first standardized to remove scale and offset differences across devices and recording conditions. Windows with extreme amplitude fluctuations, indicative of motion bursts or sensor saturation, are identified using a simple distributional check and either trimmed to remove outliers or discarded if the signal remains unstable. This step prioritizes precision over recall to ensure high-quality pretraining data.

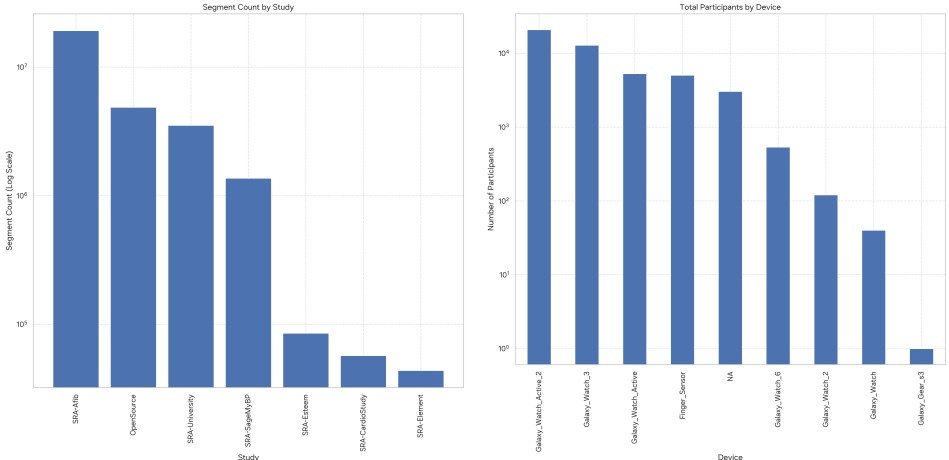

Figure 7: **Segment Count by Study.** This bar chart shows the number of data segments collected for each study, with the y-axis on a logarithmic scale to account for the large differences in segment counts.

For windows that pass amplitude screening, we assess temporal regularity by measuring short-range autocorrelation. Physiologically plausible PPG signals exhibit quasi-periodic structure; windows with highly irregular or unstable periodicity are rejected, as these patterns typically arise from motion or sensor decoupling. We additionally enforce a minimum number of cycles to eliminate degenerate or truncated traces.

Surviving windows are band-pass filtered to the cardiac frequency range to remove baseline drift and high-frequency noise while preserving pulse morphology. Signal quality is then evaluated via template matching against a canonical PPG waveform. We compute a per-window quality score that jointly reflects the fraction of the signal that matches the template and the strength of that match, penalizing cases where apparent agreement is driven by only a small portion of the window. Windows that fail this final morphology check are excluded.

This filtering is applied at scale across the corpus, retaining only windows that are clean, periodic, and morphologically consistent. The resulting pretraining set emphasizes physiologically meaningful PPG signals across devices and sampling rates, substantially reducing motion artifacts without relying on labels, heuristics tied to specific hardware, or subject-level metadata.

### D.3 DOWNSTREAM EVALUATION DATA

We evaluate HiMAE across diverse downstream tasks to assess the generality of wearable PPG representations. Rather than assuming a fixed mapping between PPG and outcomes, we exploit HiMAE's ability to learn hierarchical temporal features and adaptively resolve signal segments at scales most informative for prediction. This design allows us to probe the representational value of optical physiological signals across clinically and behaviorally relevant applications.

#### D.3.1 PVC DETECTION

Table 8: Stratified 80/20 Train/Test splits for PVC tasks (with per-task totals).

| Task | Split | Negative | Positive | Total |
|---|---|---|---|---|
| PVC Detection | train | 369987 (91.8%) | 32832 (8.2%) | 402819 |
| | test | 69880 (89.7%) | 8019 (10.3%) | 77899 |
| | totals | 439767 (91.4%) | 40950 (8.6%) | 480717 |

Premature Ventricular Contractions (PVCs) (Number Breakdowns in Table 8) are abnormal beats arising in the ventricles (Cha et al., 2012; Kaya & Pehlivan, 2015). We use paired PPG–ECG data, with ECG annotations generated using BeatLogic (Teplitzky et al., 2020) and manually verified.

PPG inputs are 10s non-overlapping wrist segments, pre-processed with a Savitzky–Golay filter (Luo et al., 2005), a 0.5–4.0 Hz bandpass, normalization to $[-1, 1]$, and exclusion of segments with motion artifacts or disruptions $> 1$ s. This task evaluates whether ubiquitous PPG can approximate arrhythmia detection typically restricted to ECG.

### D.3.2 HYPERTENSION CLASSIFICATION

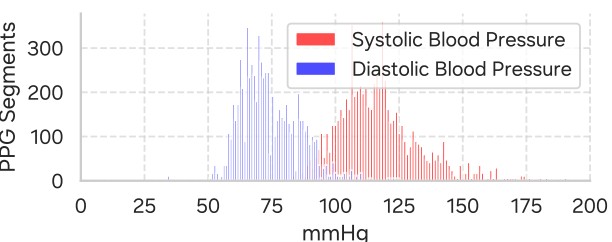

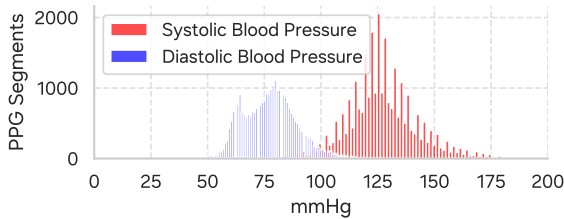

Figure 8: **Blood Pressure Distribution:** The distribution of Blood Pressure Values (mmHG) across the lab and free-living studies. We define hypertension as systolic over 130 and diastolic over 80 to generate binary outcomes.

Hypertension classification (Number Breakdowns in Figure 8) relies on cuff-based references (Simonneau et al., 2004; Giles et al., 2005; 2009; Simonneau et al., 2009; 2013; 2019). Subjects within $\pm 8$ mmHg of the diagnostic cutoff are excluded to reduce label noise, with remaining individuals labeled hypertensive or normotensive. Each 10s PPG segment undergoes Savitzky–Golay smoothing, 0.5–4.0 Hz bandpass filtering, normalization to $[-1, 1]$, and artifact removal. Unlike PVC detection, which is event-based, this task leverages PPG morphology and temporal dynamics to reflect vascular state. These evaluations contain both hypertension data collected in a naturalistic free world environment and within a controlled lab environment for both the hypertensive and blood pressure regression tasks.

### D.3.3 SLEEP STAGING

Table 9: Stratified 80/20 Train/Test splits for Sleep Staging.

| Task | Split | Wake | Light | Deep | REM | Total |
|---|---|---|---|---|---|---|
| Sleep Staging (4-class) | train | 44829 (23.9%) | 115932 (61.8%) | 6696 (3.6%) | 20214 (10.8%) | 187671 |
| | test | 11298 (23.6%) | 30153 (63.1%) | 1416 (3.0%) | 4881 (10.2%) | 47748 |
| | totals | 56127 (23.8%) | 146085 (61.9%) | 8112 (3.4%) | 25095 (10.6%) | 235419 |

Sleep staging (Number Breakdowns in Tables 9) is evaluated on the DREAMT dataset (Wang et al., 2024) hosted on PhysioNet (Goldberger et al., 2000), which includes overnight wristband data with simultaneous PSG. Annotations follow AASM standards into wake, REM, NREM1, NREM2, and NREM3, excluding missing and preparation segments. PPG is bandpass filtered (0.5–12 Hz) (Butterworth et al., 1930), segmented into 10s windows, and normalized to zero mean and unit variance. Performance is measured with five-fold subject-independent cross-validation. This task examines whether PPG encodes temporal patterns sufficient for sleep stage classification. *We note that sleep*

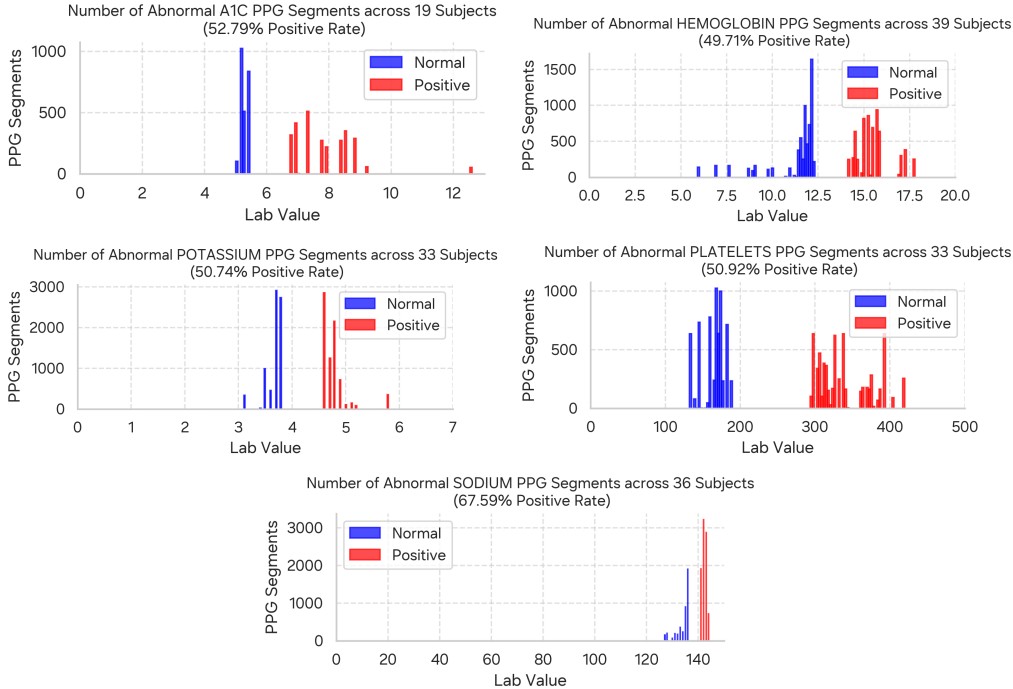

Figure 9: **Abnormal Labs Distribution:** The number of PPG segments for abnormal labs seperated based on lab-specific cutoffs. We define an abnormal lab as falling above the 75th percentile of values and a normal lab as falling within the 25th percentile.

*staging has canonically been designed by leveraging the whole sleep cycle but we are assessing the ability to monitor real time sleep staging from much shorter PPG segments.*

### D.3.4 ABNORMAL LAB TESTS

For abnormal lab test prediction, we use Samsung Galaxy Watch PPG collected at Tulane University paired with clinical laboratory results. Each test is framed as a binary classification task: outcomes are labeled negative if within the 25th percentile of lab values and the positive labels are anything above the 75th percentile (Figure 9). All other labels are excluded. Preprocessing matches other tasks. Targets include A1C, hemoglobin, platelets, potassium, and sodium, each selected for established clinical relevance. This task extends evaluation beyond cardiovascular and behavioral endpoints to systemic markers of metabolic, and hematologic health. *We note that it is unclear whether PPG can predict abnormal from healthy lab values based on the PPG alone. Despite this, Tulane univeristy presents us with an opportunity to discover if PPG signal can provide digital signatures making this an exploratory task in our benchmark.*

# E    BASELINES AND MODEL CONFIGURATION

Self Supervised Pre-trained methods have become a dominanat paradigm for health and wellness to study a variety of applications (Wornow et al., 2023; Thieme et al., 2023; He et al., 2024; An et al., 2025; Lin et al., 2025). Foundation models for one-dimensional signals are predominantly re-purposed from architectures designed for vision, with adaptations that reinterpret temporal structure as a flattened analogue of spatial correlation. In this section we highlight our baseline models and model configurations

## E.1    BASELINES

**MAE-1D** We introduce a Masked Auto ecnoder that mimics the protocol of LSM (Narayanswamy et al., 2024). This model introduces a large-scale foundation model trained on multimodal wearable sensor data but we adapt it to 1D PPG Signal. Specifically, it adopts a vision transformer archi-tecture trained via masked autoencoding with random masking. In our work, we do not replicate the full multimodal design; instead, we adapt and constrain the model to a unimodal setting for fair comparison and due to lack of open source code.

**Swin-Transformer** (Liu et al., 2021a) is a hierarchical Transformer that forms multi-scale repre-sentations by restricting self-attention to non-overlapping windows and alternating partitions with a shifted-window scheme, which enables cross-window communication while keeping computation near-linear in sequence length. We use this baseline as this is a direct comparison and counterpart to our proposed hierarchical HiMAE model. For wearable sensing, we adopt a 1D adaptation that tokenizes temporal patches and applies windowed attention along time, capturing both fine-grained waveform morphology and longer-range dependencies.

**Masked Siamese Networks** (MSN) (Assran et al., 2022) learn label-efficient representations by combining masked signal modeling with Siamese-style contrastive objectives. Instead of relying on class labels, MSN masks portions of the input and enforces consistency between augmented views. Architecturally, it employs a Vision Transformer encoder shared across views, while leveraging a predictor network to stabilize training. The key idea is to couple self-distillation with masked reconstruction to reduce sample complexity.

**DINO** (Caron et al., 2021) is a self-supervised framework that leverages knowledge distillation without labels. Using a teacher-student setup, the student network is trained to match the output distribution of the teacher under different data augmentations. Both networks are 1D-ViTs, and the method induces cluster-like emergent properties in the learned embedding space, enabling strong transfer performance without explicit contrastive pairs or handcrafted pretext tasks.

**SimCLR** (Chen et al., 2020b) establishes contrastive learning as a competitive self-supervised paradigm. The core idea is to maximize agreement between augmented views of the same sig-nal in a latent space while pushing apart representations of different images. This is implemented using a ResNET encoder (He et al., 2015), a projection head, and a contrastive loss (NT-Xent (Chen et al., 2020a)).

**PaPaGei** (Pillai et al., 2024) is a domain-specific foundation model designed for optical physiolog-ical sensing, particularly photoplethysmography (PPG). It adapts ResNET-style CNN architectures to learn robust, generalizable representations from large-scale optical physiological datasets. Pa-PaGei releases both model weights and datasets to support reproducibility and broader adoption in physiological signal analysis. In our work, we used their source code to benchmark their method by pre-training on our volume of data to ensure fair comparison.

## E.2    HYPERPARAMETERS FOR HIMAE AND BASELINES

To ensure a fair comparison across models, we aligned the training setup as closely as possible to the original implementations while maintaining consistency in optimizer choice and scheduling. All the methods trained from scratch (HiMAE, MAE-1D, Swin-Transformer, MSN, DINO, SimCLR) were trained under identical optimization regimes, while PaPaGei follows its released open source training protocol. Table 10 summarizes the key hyperparameters for all models.

Table 10: Hyperparameter Configurations for Different Models

| Configuration | HiMAE | MAE-1D | Swin-Transformer | MSN | DINO | SimCLR | PaPaGei |
|---|---|---|---|---|---|---|---|
| Training Steps | | | 50000 | | | | 15000 |
| Warmup Steps | | | 2500 | | | | — |
| Optimizer | | | AdamW (Loshchilov & Hutter (2017)) | | | | |
| Opt. momentum $[\beta_1, \beta_2]$ | [0.9, 0.95] | [0.9, 0.95] | [0.9, 0.95] | [0.9, 0.99] | [0.9, 0.99] | [0.9, 0.99] | — |
| Base learning rate | 0.001 | 0.005 | 0.005 | 0.001 | 0.004 | 0.001 | 0.0001 |
| Batch size | | | 2048 | | | | 256 |
| Weight decay | | | 0.0001 | | | | — |
| Gradient clipping | 1.0 | 1.0 | 1.0 | 3.0 | 3.0 | 3.0 | — |
| Dropout | | | 0.0 | | | | — |
| Learning rate schedule | | | Linear Warmup & Cosine Decay | | | | |
| Loss Function | | Mean Squared Error | | | Cross Entropy | Contrastive Loss | |
| Data resolution | | | 1 (signal) - 100 Hz (Sampling Rate) $\times$ 10 (seconds) | | | | |
| Augmentation | | | Flip, Time-Warping, Noise | | | | |

## E.3 LAYER WISE ANALYSIS

**Layer Wise Analysis**

```python
def layerwise_probe(model, dataloader, labels, device):
    """
    For all architectures we use the full sequence embedding across
    the temporal dimension, without collapsing to a single summary token,
    to ensure that downstream probes have access to resolution-specific
        information.
    """

    model.to(device).eval()
    hooks, acts = [], {}

    # capture encoder layer outputs
    for i, layer in enumerate(model.encoder_layers):
        hooks.append(layer.register_forward_hook(lambda m, x, y, i=i: acts
            .setdefault(i, y.detach().cpu())))

    Xs = {i: [] for i in range(len(model.encoder_layers))}
    ys = []
    for xb in dataloader:
        xb = xb.to(device)
        with torch.no_grad():
            _ = model(xb.transpose(1, 2))
        for i, a in acts.items():
            Xs[i].append(a.flatten(1).numpy()) # flatten embedding to
                 preserve time
        ys.append(labels[: xb.size(0)])
        labels = labels[xb.size(0):]

    results = {}
    for i, feats in Xs.items():
        X = np.concatenate(feats)
        y = np.concatenate(ys)
        X = StandardScaler().fit_transform(X)
        clf = LogisticRegression(max_iter=1000)
        auc = cross_val_score(clf, X, y, cv=StratifiedKFold(5), scoring="
            roc_auc").mean()
        results[f"layer_{i}"] = auc

    for h in hooks: h.remove()
    return results
```

The layer-wise analysis examines how temporal resolution in the learned representations aligns with performance on downstream tasks. For each encoder block, we extract and flatten the full sequence embedding to preserve temporal detail, allowing probes to access features at different levels of abstraction. By training logistic regression classifiers on these embeddings, we can assess which layers best capture task-relevant temporal patterns.

# F ADDITIONAL RESULTS

## F.1 MODEL CONFIGURATIONS ABLATIONS

We conducted a comprehensive ablation study of HiMAE on a 100 Hz dataset comprising ten million segments (roughly 30k hours). The experiments systematically varied architecture and hyperparameters to understand their effect on reconstruction quality (Extrapolation task from our generative benchmark in tables where it is not explicitly stated as previously done in (Narayanswamy et al., 2024)), with multiple independent training runs averaged to reduce variance from stochastic initialization and data sampling. Unless otherwise noted, all training employed AdamW with a learning rate of $3 \times 10^{-4}$, cosine decay scheduling, and a batch size of 512.

**Architecture.** We evaluated HiMAE alongside CNN baselines across increasing network depths, defined by the sequence of hidden channel dimensions $[16, 32, 64]$, $[16, 32, 64, 128]$, and $[16, 32, 64, 128, 256]$. Table 11 lists the parameter counts, showing a modest growth for HiMAE compared to CNN baselines, with the skip-connected HiMAE exhibiting slightly higher capacity than its no-skip variant.

Table 11: Model Parameters (in K or M)

| Model
*Depth* | **3-layer**
[16,32,64] | **4-layer**
[16,32,64,128] | **5-layer**
[16,32,64,128,256] |
|---|---|---|---|
| CNN | 26.2 K | 108 K | 437 K |
| HiMAE-no skip | 66.1 K | 271 K | 1.10 M |
| HiMAE | 75.3 K | 309 K | 1.25 M |

The impact of network depth on mean absolute error (MAE) and mean squared error (MSE) is summarized in Table 12. Increasing depth consistently reduced both MAE and MSE for HiMAE, with the deepest configuration yielding the lowest reconstruction error. Skip connections were critical, as HiMAE consistently outperformed its no-skip variant across all depths.

Table 12: MAE and MSE for Different Network Depths

| Model
*Depth* | **3-layer**
[16,32,64] | | **4-layer**
[16,32,64,128] | | **5-layer**
[16,32,64,128,256] | |
|---|---|---|---|---|---|---|
| | MAE ↓ | MSE ↓ | MAE ↓ | MSE ↓ | MAE ↓ | MSE ↓ |
| CNN | 0.405 | 0.234 | 0.417 | 0.249 | 0.400 | 0.231 |
| HiMAE-noskip | 0.403 | 0.236 | 0.400 | 0.246 | 0.397 | 0.233 |
| HiMAE | 0.400 | 0.230 | 0.389 | 0.223 | **0.382** | **0.221** |

**Patch Size.** We varied the spatial-temporal patch sizes over 1, 5, 10, and 20. The results in Table 14 indicate that 5 provided the best trade-off between local resolution and generative performance. Smaller patches increased flexibility but slightly degraded performance due to reduced receptive field per token, while overly large patches caused loss of fine-grained structure.

Table 14: Model Performance for Different Patch Sizes

| Model | **1** | | **5** | | **10** | | **20** | |
|---|---|---|---|---|---|---|---|---|
| | MAE ↓ | MSE ↓ | MAE ↓ | MSE ↓ | MAE ↓ | MSE ↓ | MAE ↓ | MSE ↓ |
| CNN | 0.414 | 0.239 | 0.400 | 0.231 | 0.412 | 0.244 | 0.427 | 0.261 |
| HiMAE-noskip | 0.406 | 0.239 | 0.397 | 0.233 | 0.403 | 0.246 | 0.419 | 0.262 |
| HiMAE | 0.389 | 0.226 | **0.382** | **0.221** | 0.386 | 0.231 | 0.403 | 0.247 |

**Convolution Kernel Size.** Kernel size was varied over $\{1, 5, 10, 20\}$. Table 15 shows that 5 yielded the lowest errors across all models, suggesting moderate receptive fields match the temporal and

spatial scales of our data. Very small kernels restricted context aggregation, while very large kernels oversmoothed latent features.

Table 15: Model Performance Across Convolution Kernel Sizes

| Model | 1 | | 5 | | 10 | | 20 | |
|---|---|---|---|---|---|---|---|---|
| | MAE ↓ | MSE ↓ | MAE ↓ | MSE ↓ | MAE ↓ | MSE ↓ | MAE ↓ | MSE ↓ |
| CNN | 0.416 | 0.241 | 0.401 | 0.230 | 0.410 | 0.241 | 0.424 | 0.257 |
| HiMAE-noskip | 0.409 | 0.242 | 0.395 | 0.233 | 0.403 | 0.244 | 0.420 | 0.259 |
| HiMAE | 0.392 | 0.228 | **0.382** | **0.220** | 0.388 | 0.231 | 0.404 | 0.248 |

**Stride.** We evaluated stride values of 2, 4, and 8 (Table 16). Smaller strides yielded the best performance, particularly for HiMAE, by preserving high temporal resolution in early feature maps. Performance degraded monotonically with stride increases.

Table 16: Model Performance Across Stride Values

| Model | 2 | | 4 | | 8 | |
|---|---|---|---|---|---|---|
| | MAE ↓ | MSE ↓ | MAE ↓ | MSE ↓ | MAE ↓ | MSE ↓ |
| CNN | **0.401** | **0.231** | 0.413 | 0.244 | 0.431 | 0.267 |
| HiMAE-noskip | **0.397** | **0.233** | 0.409 | 0.247 | 0.427 | 0.270 |
| HiMAE | **0.382** | **0.220** | 0.392 | 0.232 | 0.410 | 0.250 |

**Masking Ratio.** Finally, we explored the effect of varying the latent masking ratio in the masked autoencoding objective for generative tasks, with ratios from 0.5 to 0.9. As shown in Table 17, interpolation and extrapolation both improved when increasing the ratio up to 0.8, after which performance degraded for interpolation and collapsed for extrapolation.

Table 17: MAE and MSE for HiMAE Across Different Masking Ratios Evaluated on Generative Tasks

| HiMAE Masking Ratio | Temporal Interpolation | | Temporal Extrapolation | |
|---|---|---|---|---|
| | MAE ↓ | MSE ↓ | MAE ↓ | MSE ↓ |
| 0.5 | 0.397 | 0.229 | 0.407 | 0.251 |
| 0.6 | 0.388 | 0.222 | 0.397 | 0.229 |
| 0.7 | 0.384 | 0.220 | 0.396 | 0.227 |
| 0.8 | **0.379** | **0.218** | **0.387** | **0.221** |
| 0.9 | 0.381 | 0.221 | 0.288 | 0.221 |

**Final Selection.** These controlled experiments informed the final HiMAE configuration: the deepest architecture $[16, 32, 64, 128, 256]$ with skip connections, patch size 5, kernel size 5, stride 2, and a masking ratio of 0.8, which jointly achieved the best trade-off between reconstruction fidelity and parameter efficiency.

## F.2 ECG PRE-TRAINING

HiMAE attains the lowest masked-reconstruction error on ECG (Table 18), indicating that its hierarchical masking and reconstruction inductive biases capture reconstruction capacity beyond PPG. MAE-1D (ViT) is a close second, while the ablated HiMAE and CNN trail, reinforcing that the full HiMAE design transfers effectively to the ECG domain.

Table 18: Masked-reconstruction loss on ECG masked auto encoding task.

| Model | MSE (↓) |
|---|---|
| HiMAE | **0.148** |
| MAE-1D (ViT) | 0.162 |
| HiMAE (no skip) | 0.184 |
| CNN | 0.207 |

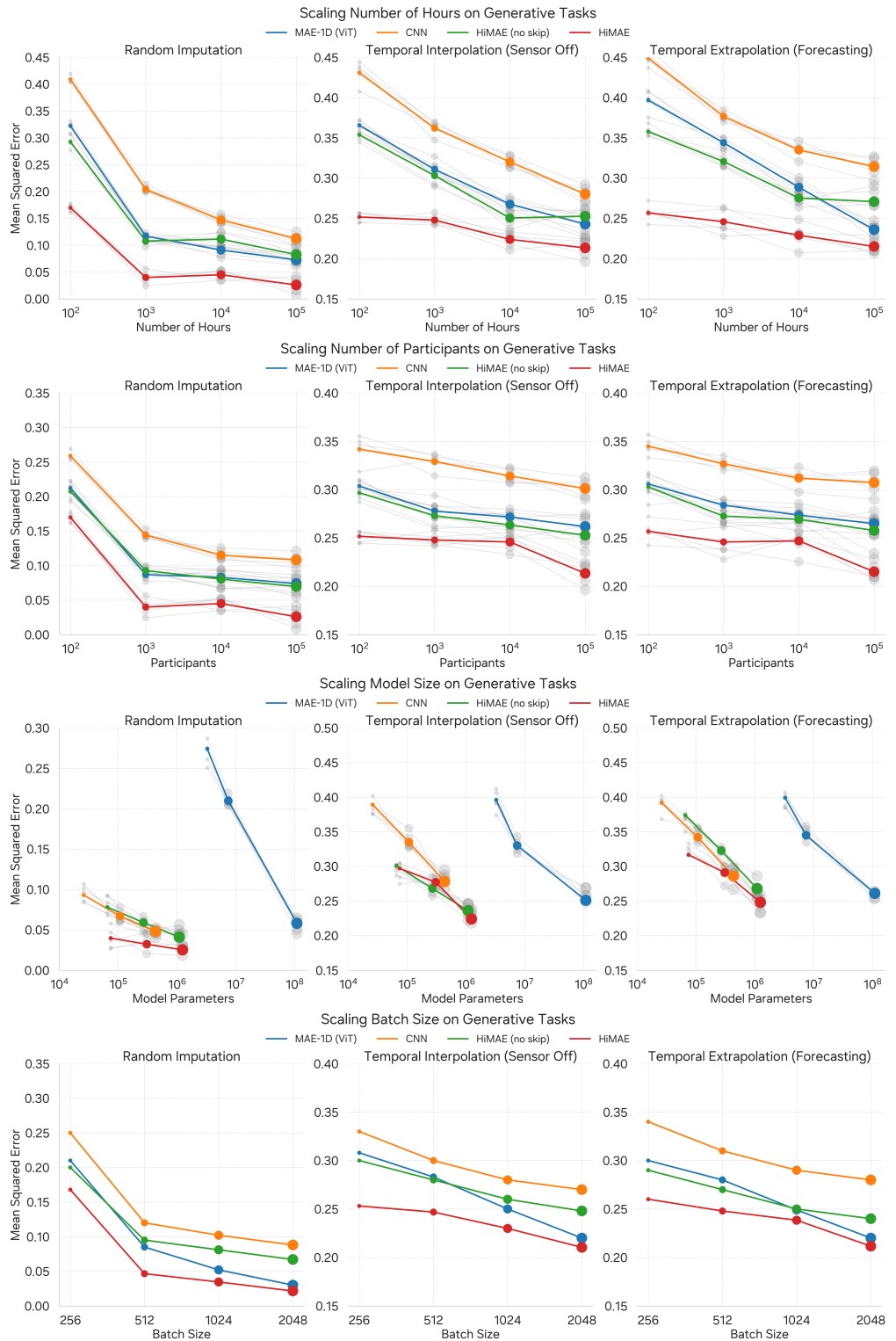

Figure 10: **Scaling Experiments on Generative Tasks:** Evaluation on the three generative tasks. HiMAE consistenly outperforms all model at our scale of data

### F.3 SCALING RESULTS FOR GENERATIVE TASKS

**Scaling analysis.** We evaluate HiMAE's reconstruction error under participant, recording hour, batch size, and model size scaling, following the regimes of Narayanswamy et al. (2024); Xu et al. (2025): random imputation, temporal interpolation, and temporal extrapolation. Across all settings HiMAE follows clean scaling law trends (Kaplan et al., 2020) and maintains a margin over MAE-1D (ViT) and CNN baselines.

The most pronounced effect is model size. At small capacities HiMAE achieves lower error than much larger transformer baselines, highlighting the advantage of hierarchical inductive bias over sheer parameter count. MAE-1D only begins to close the gap at orders of magnitude more parameters. The transformer could surpass our HiMAE model when given a larger capacity but this again highlights the effectiveness of the inductive bias that we are conveying. Participant, hour, and batch size scaling follow canonical patterns. More participants and longer recordings steadily reduce error, with HiMAE continuing to improve where baselines saturate, especially on interpolation and extrapolation.

### F.4 HIERARCHAL CONCORDANCE

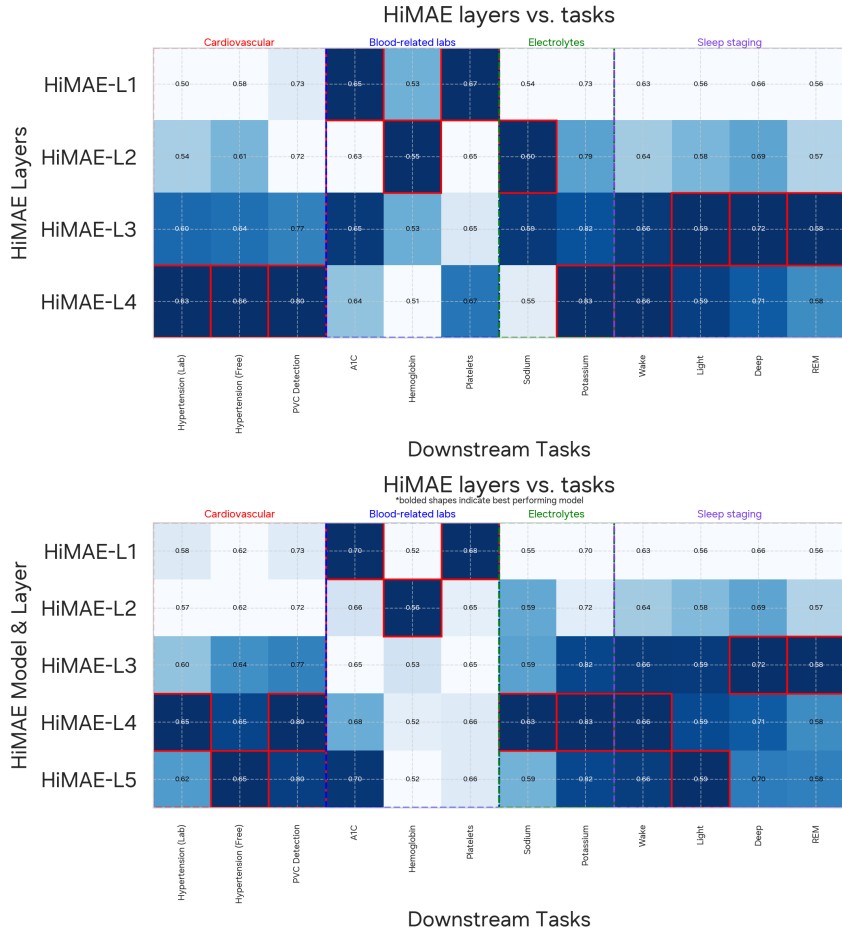

Figure 11: **HiMAE layer concordance across encoder depths.** Heatmaps compare downstream AUROC when probing HiMAE at 4 layers (top) versus 5 layers (bottom). Despite the removal of an encoder–decoder stage, the resolution–task alignment remains highly concordant: tasks such as PVC detection and hypertension consistently peak at similar layers, while sleep staging benefits from coarser representations. Minor deviations appear in intermediate layers, but the overall hierarchy of predictive resolutions is preserved, indicating robustness of the resolution hypothesis to architectural depth.

**Layer concordance across depths.** We further assess the stability of the resolution hypothesis by comparing HiMAE trained with four versus five encoder–decoder stages (Figure 11). The resulting heatmaps reveal that the alignment between downstream tasks and temporal resolutions is largely preserved across depths. Cardiovascular endpoints such as PVC detection and hypertension consistently achieve their best performance at finer layers, while blood related labs benefits from coarser layers. Although minor fluctuations appear in intermediate levels, the overall hierarchy of predictive resolutions is concordant. This suggests that the resolution–task mapping uncovered by HiMAE is not an artifact of architectural depth, but a robust property of the representations themselves.

## F.5 TRANSFORMER LAYER INTERPRETABILITY

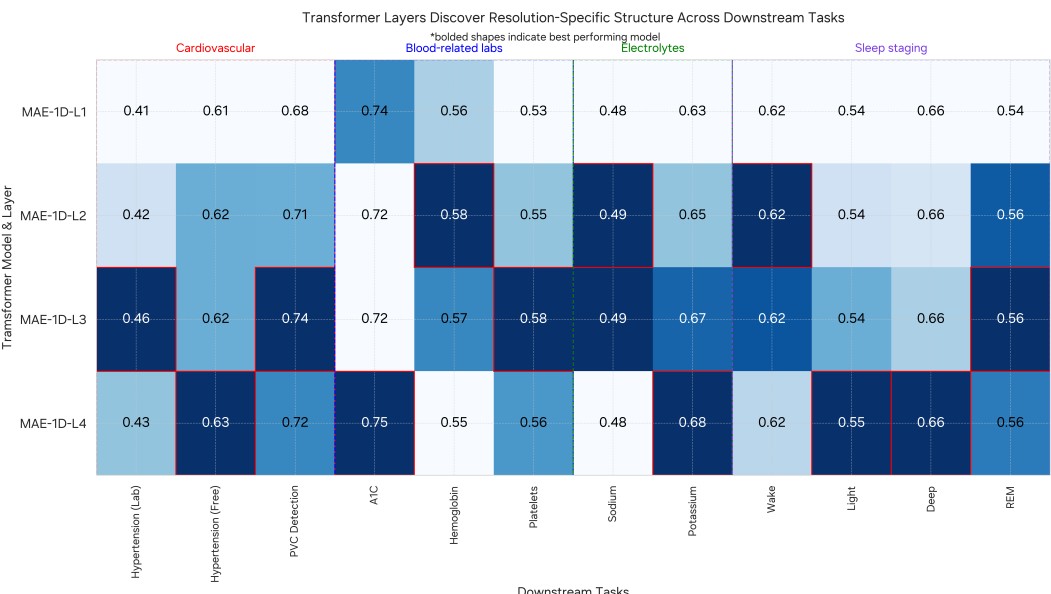

Figure 12: **Transformer based Layer Wise Analysis:** Unlike HiMAE, MAE-1D exhibits non-monotonic trends and lacks concordance with HiMAE's internal representation hierarchy. We observe that performance typically peaks in intermediate and task-specific later layers, though this pattern carries important nuance.

Figure 12 outline a deep conceptual contrast between HiMAE's layer wise interpretability from Figure 13 and Appendix Section F.4 and Transformer based encoders like MAE-1D (ViT) in how their internal representations evolve and why their layerwise "probing" behaviors differ.

In a U-Net with convolution operators, each layer has a localized receptive field that gradually expands with depth. Convolution is a local operator, so early layers capture fine-grained spatial details (edges, textures), mid layers combine local motifs into parts, and deeper layers encode semantic abstractions or whole objects. Skip connections reintroduce lost resolution but don't globalize information. This creates a hierarchical representation: the notion of "scale" is physically encoded in the architecture, with clear separations between low-level and high-level representations. When you probe features layer by layer, you observe clean transitions.

Transformers, on the other hand, start with global receptive fields from the very first layer because self-attention mixes information across all positions in one step. Every token can, in principle, interact with every other token regardless of spatial proximity. Depth therefore does not correspond to expanding spatial scale but instead refines representations through repeated global mixing and specialization. Each layer develops different attention patterns and specialized circuits (like induction heads or copy-suppression heads), rather than encoding progressively larger spatial features. Depth adds precision and compression, not hierarchical abstraction.

This makes transformer probing results very different. Since there is no strict bottom-to-top feature pyramid, probes do not show a monotonic increase in semantic abstraction. Instead, they show nonmonotonic behavior: mid layers often peak in semantic information, and late layers compress

features toward task-specific prediction spaces. Representations are distributed, overlapping, and non-hierarchical—each layer contributes globally but in subtly different ways.

Mathematically, this difference arises because convolution enforces spatial locality and translation equivariance via weight sharing and limited receptive fields, while self-attention defines a global kernel $A = \text{softmax}(QK^\top/\sqrt{d})$ that mixes all spatial positions. Hence, the "hierarchy" in CNNs is an emergent geometric property of the convolution operator, whereas in transformers the representational geometry is depth-wise iterative refinement within a globally connected graph.

## F.6   T-SNE VISUALIZATION AND REPRESENTATION INTERPRETABILITY

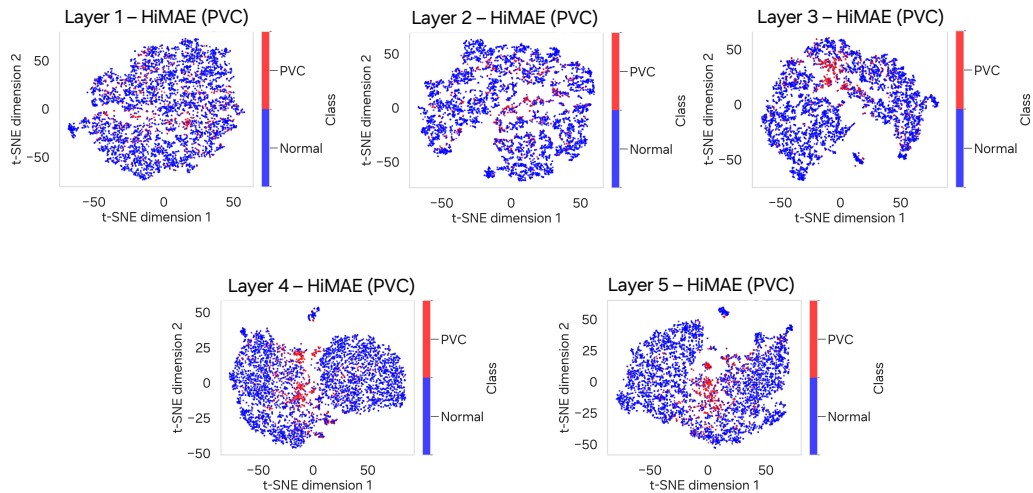

Figure 13: **t-SNE Visualization and Representation Interpretability.** We explore how the layer wise TSNE evolve on the PVC Classification task to help us understand how the representations are organized as they traverse the multiple encoders.

To characterize how HiMAE's hierarchical representations evolve across depth, we visualize t-SNE projections of encoder embeddings from each layer on the PVC detection task in Figure 13. The corresponding AUROC scores (L1: 0.73, L2: 0.72, L3: 0.77, L4: 0.80, L5: 0.80; Figure 4) closely mirror the progressive emergence of class separability observed across layers.

Representations from early layers (L1–L2) form diffuse clusters with limited separation between normal and PVC segments, indicating that these layers predominantly encode local waveform morphology. At intermediate depth (L3), distinct PVC clusters begin to emerge, coinciding with the first substantial increase in AUROC and marking a transition toward rhythm-level abstraction. Deeper layers (L4–L5) exhibit compact, well-separated clusters, consistent with representations that integrate longer temporal context and capture higher-level cardiac dynamics relevant for arrhythmia discrimination.

Together, these visualizations provide an interpretable view of HiMAE's hierarchical inductive bias: representations progressively abstract temporal information from fine-grained morphology to broader physiological context. The layer-wise evolution of t-SNE structure offers empirical support for the resolution hypothesis, suggesting that higher layers encode slower, more discriminative temporal processes that ultimately drive clinical performance.

# G ON-DEVICE EXPERIMENTS

## G.1 EXPERIMENTAL PROTOCOL

We evaluated the on-device performance of HiMAE using a Samsung Galaxy Watch 8 running Wear OS. All experiments were performed natively on-device to capture realistic latency and throughput characteristics under wearable hardware constraints. The model was deployed using PyTorch Mobile with TorchScript conversion to minimize runtime overhead and ensure compatibility with ARM-based computation. The device is powered by the Exynos W1000 chipset, featuring a 5-core CPU (1× Cortex-A78, 3× Cortex-A55, 1× Cortex-M55) fabricated on a 3 nm GAA process, and equipped with 2 GB of LPDDR5 memory.

Inference was performed at a fixed batch size of 1, corresponding to a 10-second physiological signal window sampled at 100 Hz. To ensure measurement stability, we used 20 warm-up runs followed by 100 timed inference passes. Latency was defined as the mean per-sample forward-pass time, with additional reporting of median and 95th percentile values to capture tail latencies. Throughput was defined as the total number of samples processed per second over a 10-second rolling interval.

For timing measurements, we used CUDA event synchronization on GPU and Python's high-resolution wall-clock timers on CPU. All computations were executed using float32 precision. The benchmarking routines are provided in the accompanying code listings, which include throughput and latency measurement functions.

As a point of reference, we additionally report datacenter-grade inference metrics obtained using an NVIDIA T4 GPU to contextualize the mobile device performance. Although the T4 operates at higher power, modern mobile GPUs (e.g., Qualcomm Adreno 750) demonstrate comparable inference efficiency per watt (Buber & Banu, 2018; Wesolowski et al., 2021), validating the relevance of on-device inference as a proxy for real-world deployment on consumer hardware.

**Throughput Code**

```python
def measure_throughput(model, dummy_input, device, num_seconds=10):
    """Measures inference throughput (samples/sec) with a fixed batch
        size of N."""

    model.eval()
    batch_size = N
    dummy_input = dummy_input.to(device)
    model.to(device)

    with torch.no_grad():
        for _ in range(10):
            _ = model(dummy_input)
    if device.type == 'cuda':
        torch.cuda.synchronize()

    num_inferences = 0
    start_time = time.time()
    with torch.no_grad():
        while time.time() - start_time < num_seconds:
            _ = model(dummy_input)
            if device.type == 'cuda':
                torch.cuda.synchronize()
            num_inferences += 1

    total_time = time.time() - start_time
    throughput = num_inferences * batch_size / total_time
    return {"Throughput_samples_per_sec": throughput}
```

**Latency Code (Batch Size = 1)**

```python
def measure_inference_time_bs1(model, dummy_input, device):
    """Measures inference latency (mean, median, 95th percentile) with
        a fixed batch size of 1."""
    if dummy_input.shape[0] != 1:
        raise ValueError(f"Input batch size must be 1 for this
            function. Got {dummy_input.shape[0]}")

    model.eval()
    dummy_input = dummy_input.to(device)
    model.to(device)

    with torch.no_grad():
        for _ in range(warmup_runs):
            _ = model(dummy_input)
    if device.type == 'cuda':
        torch.cuda.synchronize()

    timings = []
    with torch.no_grad():
        for _ in range(num_runs):
            if device.type == 'cuda':
                start_event = torch.cuda.Event(enable_timing=True)
                end_event = torch.cuda.Event(enable_timing=True)
                start_event.record()
                _ = model(dummy_input)
                end_event.record()
                torch.cuda.synchronize()
                elapsed_time_ms = start_event.elapsed_time(end_event)
            else:
                start_time = time.time()
                _ = model(dummy_input)
                end_time = time.time()
                elapsed_time_ms = (end_time - start_time) * 1000
            timings.append(elapsed_time_ms)

    mean_latency = np.mean(timings)
    median_latency = np.median(timings)
    std_latency = np.std(timings)
    p95_latency = np.percentile(timings, 95)
    return {
        "Mean_Latency_ms": mean_latency,
        "Median_Latency_ms": median_latency,
        "P95_Latency_ms": p95_latency,
    }
```

## G.2 INFERENCE EFFICIENCY

We benchmarked the inference efficiency of our proposed HiMAE against the transformer baseline (MAE-1D), measuring three aspects: model footprint and computational complexity in terms of parameters, memory, and FLOPs per 10-second input window at 100 Hz (Table 19); latency, defined as mean per-sample forward-pass time at batch size 1; and throughput, defined as the maximum number of samples processed per second (Table 20).

**Results** Despite being more than two orders of magnitude smaller in parameter count, the HiMAE consistently outperforms the transformer baseline across all efficiency metrics. Between Efficient-Net (Tan & Le, 2020), it remains marginally better which is encouraging due to the optimizations designed in this model.

| Model | Params | FLOPs | Memory |
|---|---|---|---|
| HiMAE | 1.2M | 0.0647 gFLOPS | 4.8 MB |
| Efficient-Net | 7.8M | 0.70 gFLOPS | 31.1 MB |
| Swin-Transformer | 110.6M | 11.89 gFLOPS | 423.8 MB |
| MAE-1D | 110.6M | 15.94 gFLOPS | 441.3 MB |

Table 19: **HiMAE is lightweight and efficient:** Model size and compute cost comparison between HiMAE and MAE-1D. FLOPs measured per forward pass on a 10s sequence at 100Hz.

| Model | GPU Lat. | GPU Thr. | CPU Lat. | CPU Thr. |
|---|---|---|---|---|
| HiMAE | 0.039 ms | 25.8k/s | 0.99 ms | 1.2k/s |
| Efficient-Net | 0.082 ms | 12.2k/s | 1.42 ms | 0.704k/s |
| Swin-Transformer | 0.704 ms | 1.42k/s | 2.95 ms | 0.456k/s |
| MAE-1D | 0.80 ms | 1.24k/s | 3.36 ms | 0.298k/s |

Table 20: **Inference Performance:** Latency (ms per sample, batch size 2048) and throughput (samples/sec) measured over 10 s windows.

**Model footprint:** HiMAE reduces parameters from 110M to 0.31M ($\sim 355\times$ fewer), FLOPs from 15.94G to 0.0647G ($\sim 246\times$ fewer), and memory from 441.3MB to 3.6MB ($\sim 123\times$ smaller). These reductions highlight that computational savings scale with the compactness of the model, without loss of representational capacity for the task.

**Latency:** HiMAE achieves substantially faster per-sample inference. On GPU, latency drops from 0.80ms to 0.039ms ($\sim 20\times$ faster), while on CPU it falls from 3.93ms to 0.99ms ($\sim 4\times$ faster). The reduction in latency follows directly from the smaller computational footprint, reflecting a consistent efficiency advantage.

**Throughput:** These improvements translate into higher throughput across hardware. On GPU, throughput increases from 1.24k to 25.8k samples/s ($\sim 21\times$ higher), while CPU throughput rises from 0.255k to 1.2k samples/s ($\sim 5\times$ higher). These results confirm that computational gains extend beyond memory and FLOPs, yielding end-to-end speedups at inference time.

In summary, HiMAE achieves a favorable tradeoff between compactness and efficiency, providing lower FLOPs, smaller memory footprint, and faster inference despite its reduced model size. It also outperforms Efficient-Net B1 which was specially designed and optimized for performance and compactness giving a comparison and context to our models performance.

## H  FREQUENTLY ASKED QUESTIONS

**What are the main conclusions from this work?** We demonstrate that convolutional architectures benefit from inductive biases that remain advantageous for PPG signals. On our pre-training data, our model consistently outperforms alternative baselines. Furthermore, scaling experiments across model sizes reveal that brute-force scaling of generic architectures is possible, but less effective: our model achieves stronger performance and scales more gracefully due to a better initialization and inductive structure relative to other models. In addition to this inductive bias and compact design, our contributions are two fold in the sense that our model demonstrates the first on-device model which does not require phone level processors to run inference.

**Is your pre-training dataset large enough?** Our pre-training corpus was collected internally and is of comparable scale to recent public benchmarks such as PaPaGei and Apple's datasets. In terms of magnitude, we position our dataset as PaPaGei (Pillai et al., 2025) $<$ Ours $<$ Apple (Abbaspourazad et al., 2023) $<$ Google (Narayanswamy et al., 2024). Thus, while not the largest available, our dataset size is sufficiently large to validate the approach and lies within the range of accepted practice for self supervised learning wearable models.

**Why do you model at 10-second windows?** We deliberately adopt 10s windows sampled at 100Hz to balance physiological coverage with on-device feasibility. Many clinically actionable events, such as arrhythmic beats or premature ventricular contractions, unfold on the order of seconds and require rapid detection to enable continuous monitoring and real-time feedback. Shorter windows would impair the model's ability to capture meaningful temporal context, while much longer windows would hinder low-latency inference on watch-class hardware. By constraining the receptive field to 10s, HiMAE preserves second-level resolution while remaining efficient enough to process signals continuously under the hardware limits of edge devices. Additionally, 10-second window are a standard protocol that are adopted in the clinical setting where ECG for example is collected and interpreted at 10 second segments (Shuai et al., 2016).

**What are the advantages of smaller models?** From a research perspective, smaller models foster inclusivity by reducing reliance on brute-force scaling of transformer-based architectures that only industry-scale labs can realistically afford. From a deployment standpoint, compact models enable on-device inference on constrained hardware such as wearables. This dual benefit—lower research barriers and wider deployment potential—underscores the importance of investigating architectures that remain competitive at modest scale.

**How large is too large to deploy on a smart watch?** In principle, models up to approximately 50MB can be stored and executed on modern smart watches or larger models can be quantized (Jacob et al., 2017). In practice, however, latency and energy considerations suggest that models exceeding roughly 10MB may already hinder real-time inference and limit commercial viability. Additionally quantization does not do due dilligence to the original model and some level of the model's performance is lost. While smartphones relax these constraints, our contribution highlights that the proposed model remains sufficiently compact to fit within the computational and storage budgets of wearable devices such as watches, thereby supporting direct on-device deployment.

**Can PPG predict abnormal laboratory results?** We frame this as a binary classification task, testing whether photoplethysmography signal encodes biomarkers that separate "normal" from "abnormal" lab classes. Our investigation probes whether learned PPG representations capture biomarker signatures correlated with out-of-range labs, using lightweight classifiers on frozen embeddings with strict temporal alignment. Preliminary evidence suggests discriminative signal above chance, but these findings are designed to be exploratory and not clinically actionable.

