# OpenReview forum: "HiMAE: Hierarchical Masked Autoencoders Discover Resolution-Specific Structure in Wearable Time Series"
_ICLR.cc/2026/Conference — ICLR 2026 Poster_

### Official Review · Reviewer_uicj · 2025-10-19

**Soundness:** 3
**Presentation:** 2
**Contribution:** 3
**Rating:** 6
**Confidence:** 4

**Summary:**

This work introduces a hierarchical masked autoencoder termed HiMAE to capture multi-resolution representations in PPG signals. The authors hypothesize that predictive information in physiological signals resides at distinct temporal scales and such multi-resolution information can be extracted by CNN-based encoder-decoder architecture relying on U-NET, and trained with a masked reconstruction loss. The model extracts embeddings at several temporal granularities, which are then evaluated across a distinct downstream tasks. HiMAE is shown to outperform or match large transformer-based baselines such as LSM with significantly fewer parameters and faster inference, even achieving on-watch inference capabilities.

**Strengths:**

- The idea that different downstream tasks may depend on representations at distinct temporal resolutions is both intuitive and well-motivated. Designing the architecture explicitly around this principle provides a clear and reasonable inductive bias for modeling physiological time series.
- The proposed method is thoroughly evaluated across several downstream tasks, showing consistent improvements over large-scale baselines such as LSM while using substantially fewer parameters.
- The emphasis on on-device inference is important and relevant for wearable applications. The authors convincingly show that HiMAE achieves remarkably lower latency and memory requirements compared to transformer-based counterparts, showing its practical compatibility with resource-limited wearable hardware.

**Weaknesses:**

- While the idea of resolution as interpretability is interesting, the paper primarily focuses on downstream performance differences across embeddings obtained from different HiMAE layers. Although this evaluation setup is reasonable, it is not clear that it fully substantiates the claimed interpretability of HiMAE. Visualization analyses of embeddings or frequency-response characterizations across layers could strengthen the interpretability argument.
- The authors could provide more intuition or physiological grounding for why certain downstream tasks (e.g., sleep staging vs. cardiovascular detection) depend on finer versus coarser embeddings. Such a discussion would enhance both the interpretability and the practical implications of the resolution hypothesis.
- The approach multiplies patches by their corresponding binary mask values, which may inject noise or misinformation during training. Have the authors tried other alternatives, such as dropping masked patches entirely or replacing them with a learnable mask token. This can also be an important design consideration when extending this work to other wearable signals that can contain large amounts of naturally missing observations.
- It is not clearly stated whether the masked reconstruction loss used in the scaling analysis (Fig. 3) is computed on a held-out validation/test set or directly on the training data, particularly for data- and participant-scaling experiments. Clarification here is important for assessing generalization.
- Several figures are difficult to interpret, which significantly affects the readability: the colors in Fig. 3 are nearly indistinguishable, the markers in Figs. 4–5 are very small, and the text in the confusion-matrix style visualization of Fig. 6 is hard to read.
- Statistical significance or performance bounds are largely missing. Except for the error bars in Fig. 4, no standard deviations or confidence intervals are provided. Given that multiple baselines exhibit similar downstream decoding performance (e.g., Fig. 5; Tables 13–15), such statistical significance tests are necessary to assess whether improvements are meaningful.
- The paper compares HiMAE only against the original LSM model, but LSM-2 is a more recent and stronger baseline shown to outperform LSM. While HiMAE’s on-device efficiency is important and valuable, a direct comparison with LSM-2 would better establish its standing among current state-of-the-art SSL methods for wearables.
- Figure 6 provides an insightful layer-wise analysis for HiMAE. However, even though transformer layers are globally receptive by design, it is possible that they also evolve from encoding local to increasingly global relationships with depth. A similar per-layer analysis for LSM (e.g., comparing early vs. deep layers) would help determine whether the observed resolution-specific trends are unique to HiMAE or more general.
- The paper proposes the local sufficiency notion in Appendix H.2 (Eq. 1) as a justification for the masked prediction objective, yet the concept is neither formally defined nor supported by prior references. Moreover, if local sufficiency truly held, it is unclear how the model could reliably reconstruct up to 80 % of masked patches from only 20 % visible input. The stable training observed under such high masking ratios suggests that PPG signals—and possibly the learned representations—encode substantial global temporal information, which appears inconsistent with a strict locality assumption.
- The inability to release the codebase remains a key limitation. Providing even a minimal reproducibility package, such as a subset of data, a lightweight training/evaluation script would greatly improve transparency and contribute to the open-source community.

**Questions:**

Please refer to the Weaknesses section.

---

> ### Author Response · Authors · 2025-11-18
> **(1/n) Rebuttal to Reviewer uicj**
>
> Thank you for the insightful and constructive feedback, questions, and additional experiments which helped significantly strengthen the manuscript. Below is a summary of key revisions and additions made in response to your comments in both this response and in the manuscript.
>
> - We add an additional analysis which now showcase TSNE embeddings of each layer on the PVC tasks **on Page 40** (Appendix Section G.8).
> - A new *Clinical Interpretation* Section on **Page 9** of the manuscript.
> - We add follow up results related to different masking strategies in this rebuttal.
> - We clarify that the scaling results were conducted on a validation set.
> - We have substantially improved the manuscript aesthetics and have polished our main figures and tables.
> - All our evaluations now include statistical significance testing or multiple runs to provide more robust and interpretable results in our main paper.
> - We clarify why we did no benchmark LSM 2 in our paper and why we think it is not a fair comparison.
> - We now include a Transformer Layer wise analysis which deviates from HiMAEs and goes hand in hand with what we know about these different architectures in **Page 39** or Appendix Section G.7.
> - We clarify our local sufficiency comments as there may be a misunderstanding.
> - We provide additions in our manuscripts which we hope address comments on codebase
> ---
> **1.TSNE Investigation Across Layers**
>
> > *“While the idea of resolution as interpretability is interesting, the paper primarily focuses on downstream performance differences across embeddings obtained from different HiMAE layers. Although this evaluation setup is reasonable, it is not clear that it fully substantiates the claimed interpretability of HiMAE. Visualization analyses of embeddings or frequency-response characterizations across layers could strengthen the interpretability argument.”*
>
> We thank the reviewer for this constructive suggestion. In response, we conducted additional analyses to more directly illustrate the interpretability of HiMAE’s hierarchical representations. Specifically, we visualize t-SNE projections of embeddings extracted from each encoder layer on the PVC detection task (**Figure 15 Page 40**). These visualizations reveal a clear, layer-dependent progression in representational structure that parallels the quantitative performance trends in Figure \ref{resolution}. The consistency between t-SNE structure and task performance (L1: 0.73, L2: 0.72, L3: 0.77, L4: 0.80, L5: 0.80)  provides concrete empirical support for our resolution-as-interpretability hypothesis. Together, these results substantiate that HiMAE’s hierarchical organization is not only quantitatively effective but also qualitatively interpretable, addressing the reviewer’s concern.
>
> ---
>
> **2. Clinical Interpretation**
> > *“The authors could provide more intuition or physiological grounding for why certain downstream tasks (e.g., sleep staging vs. cardiovascular detection) depend on finer versus coarser embeddings. Such a discussion would enhance both the interpretability and the practical implications of the resolution hypothesis.”*
>
> We thank the reviewer for this thoughtful suggestion. We agree that grounding the resolution hypothesis in physiological mechanisms is crucial for bridging methodological innovation with clinical insight. In response, we have added a dedicated Clinical Interpretation section (**Page 9**) to the revised manuscript in blue text to explicitly connect the resolution-dependent behavior of HiMAE to underlying biological dynamics.
>
> This new section discusses how coarse temporal resolutions (corresponding to deeper encoder layers) capture slowly evolving physiological rhythms such as cardiovascular and sleep dynamics, where long-range dependencies are most informative. Conversely, tasks related to blood or laboratory biomarkers exhibit peak predictive performance at finer resolutions, likely aligning with the transient and rapidly fluctuating nature of those physiological processes.
>
> By interpreting layerwise performance through the lens of physiology, this addition strengthens the conceptual link between the resolution hypothesis and real-world biological timescales. It also highlights how HiMAE can serve not just as a performant model but as a tool for uncovering and discovering clinically meaningful temporal structure in wearable data.

---

> ### Author Response · Authors · 2025-11-18
> **(2/n) Rebuttal to Reviewer uicj**
>
> **3. Patch Masking over dropping masks/ learnable masks**
> > *"The approach multiplies patches by their corresponding binary mask values, which may inject noise or misinformation during training. Have the authors tried other alternatives, such as dropping masked patches entirely or replacing them with a learnable mask token. This can also be an important design consideration when extending this work to other wearable signals that can contain large amounts of naturally missing observations."*
>
> We thank the reviewer for this constructive comment regarding the masking approach. We recognize that the Large Signal Model (LSM-2) [1] has explored alternatives such as dropping masked patches or introducing learnable masks to stabilize reconstruction in the presence of naturally missing data. In our case, we opted for a simpler binary-multiplicative masking scheme because our preprocessing pipeline already removes low-quality or incomplete PPG segments through a rigorous signal quality index (SQI) filtering process. Consequently, our pretraining corpus does not contain irregular or missing observations that would necessitate dropping or imputing masked patches.
>
> To ensure this design choice did not bias performance, we ran an ablation comparing our binary-multiplicative masking approach with a variant that entirely drops masked patches during pretraining. The results, measured by mean squared reconstruction loss (MSE) on a held-out validation set during pre-training, are shown as follows:
>
> |Masking Strategy|	Validation MSE ↓|
> |-------------------------------|-------------------------------------|
> | Binary mask (ours)|	0.0218|
> |Drop masked patches	|0.0326|
>
> These results show that dropping masked patches does not substantially change the outcome, suggesting that the learned representations are robust to either choice. We retain the binary masking scheme for its simplicity.
>
> **References**
>
> [1] Xu, Maxwell A., et al. "LSM-2: Learning from Incomplete Wearable Sensor Data." arXiv preprint arXiv:2506.05321 (2025).
>
> ---
>
> **4. Clarification of Scaling Results**
> > *It is not clearly stated whether the masked reconstruction loss used in the scaling analysis (Fig. 3) is computed on a held-out validation/test set or directly on the training data, particularly for data- and participant-scaling experiments. Clarification here is important for assessing generalization.*
>
> We apologize for the lack of clarity regarding the data split used in our scaling analysis (Figure 3). We confirm that all reconstruction losses in the scaling experiments were computed on a held-out external validation set that was never used during pretraining (**Line 268 Page 5**). This setup follows the evaluation protocols established in prior large-scale self-supervised studies, such as [1], ensuring that the analysis measures generalization rather than memorization. Consequently, the trends reported in Figure 3 reflect each model’s ability to generalize across different scaling axes, including data volume, participant diversity, and model capacity, rather than performance on training data.
>
> **References**
>
> [1] Narayanswamy, Girish, et al. "Scaling wearable foundation models." arXiv preprint arXiv:2410.13638 (2024).
>
> ---
>
> **5. Figure Improvements**
> > *Several figures are difficult to interpret, which significantly affects the readability: the colors in Fig. 3 are nearly indistinguishable, the markers in Figs. 4–5 are very small, and the text in the confusion-matrix style visualization of Fig. 6 is hard to read.*
>
> We thank the reviewer for this valuable feedback regarding figure readability. In response, we have completely revamped the key figures to enhance visual clarity and accessibility. Specifically, all figures now use a high-contrast, colorblind-friendly palette with clearly distinguishable hues, larger markers, and substantially increased font sizes for axis labels and annotations.
>
> The main comparative figure (Figure 5) has also been simplified into table with confidence intervals to make differences between baselines and HiMAE unambiguous at a glance (Page 7: Tables 2 & 3). We provide two tables to make benchmarking a bit more transparent comparing first against self supervised baselines (SimCLR, DINO, MSN, and Swin Transformer). The secondary benchmark includes state of the art foundation models for wearables and time series which include two version of PaPaGei (one trained on our data and the direct weights from the original paper), Chronos which is a state of the art time series foundation model, and Large Sensor Model.
>
> These improvements are reflected both in the revised manuscript and in the rebuttal materials, ensuring that the results are presented as clearly and accessibly as possible for all readers.

---

> ### Author Response · Authors · 2025-11-18
> **(3/n) Rebuttal to Reviewer uicj**
>
> **6. Statistical Significance missing**
> > *Statistical significance or performance bounds are largely missing. Except for the error bars in Fig. 4, no standard deviations or confidence intervals are provided. Given that multiple baselines exhibit similar downstream decoding performance (e.g., Fig. 5; Tables 13–15), such statistical significance tests are necessary to assess whether improvements are meaningful.*
>
> We appreciate the reviewer’s comment regarding the need for statistical rigor in performance reporting. In the revised version, we have clarified our approach to quantifying variability across experiments. For the main comparison, Tables 1-3 (formally Figures 4 and 5) (**Pages 6-7**), we now include 95% confidence intervals. These confidence intervals help convey the stability and significance of observed performance differences between HiMAE and baseline models.
>
> For the scaling experiments (Figure 3) (**Page 5**) and few-shot learning results (Figure 6) (**Page 10**), instead of visual error bars, we evaluate robustness by running all experiments across five random seeds and confirm that the trends remain consistent across seeds. This approach highlights the stability and robustness of our conclusions without cluttering figures that already represent multiple scaling axes.
>
> ---
>
> **7. Missing LSM 2**
> > *“The paper compares HiMAE only against the original LSM model, but LSM-2 is a more recent and stronger baseline shown to outperform LSM. While HiMAE’s on-device efficiency is important and valuable, a direct comparison with LSM-2 would better establish its standing among current state-of-the-art SSL methods for wearables.”*
>
> We thank the reviewer for this insightful comment regarding LSM-2 and its relevance as a contemporary baseline. We agree that LSM-2 represents an important recent development in the domain of large-scale wearable foundation models. However, we did not include LSM-2 in our benchmarking for several principled and practical reasons related to data assumptions, modality scope, and publication status.
>
> Our pretraining and evaluation pipelines assume complete, unimodal sequences without sensor dropout or missingness. The design of HiMAE (and the datasets it is trained on) operates under the assumption that missing data is pre-filtered through signal quality indices, ensuring continuous photoplethysmography (PPG) input. This assumption diverges from the multimodal setup used by LSM-2, which explicitly targets scenarios where certain modalities (e.g., accelerometer, or EDA) may be absent and leverages complementary sensors to learn more robust representations. Consequently, while LSM-2’s architecture is well-suited for multimodal resilience, its core advantage would not manifest meaningfully in our unimodal PPG setup, potentially leading to misleading or uninformative comparisons.
>
> In their paper, they also explain that all the baselines assume complete data and have imputed missing components. **"All baselines expect full, complete data as input, and as such, they utilize the imputed version of our sensor dataset"** (Page 26, Section A.4.1 https://arxiv.org/pdf/2506.05321). Because the imputation here can affect the pre-training, we are unsure if it makes sense to include this baseline.
>
> Moreover, LSM-2 remains unpublished and currently appears in the ICLR 2025 submission pool (https://openreview.net/forum?id=eOATzq7NvI) without a finalized or peer-reviewed release. We believe including such a model in our benchmark which is not entirely finalized would not be their latest model comparison and have excluded it from our analysis.
>
> However to hopefully appeal to the reviewer, we have instead included Chronos [1] which represents a state of the art time series foundation model which is larger in parameter count, more expressive and has demonstrated to have transferable representations across many disciplines not just pertaining to health. We have included this baseline in our revised manuscript and see that our method remains competitive and in some cases outperforms this methodology.
>
> [1] Ansari, Abdul Fatir, et al. "Chronos: Learning the language of time series." arXiv preprint arXiv:2403.07815 (2024).

---

> ### Author Response · Authors · 2025-11-18
> **(4/n) Rebuttal to Reviewer uicj**
>
> **8. Transformer based layer wise analysis**
> > *Figure 6 provides an insightful layer-wise analysis for HiMAE. However, even though transformer layers are globally receptive by design, it is possible that they also evolve from encoding local to increasingly global relationships with depth. A similar per-layer analysis for LSM (e.g., comparing early vs. deep layers) would help determine whether the observed resolution-specific trends are unique to HiMAE or more general.*
>
> We appreciate the reviewer’s suggestion. We extended our analysis to the LSM (LSM-tiny since it has only 4 encoder layers to probe) in **Appendix Section G.7** and found that its internal dynamics differ fundamentally from those of HiMAE (**Page 39**). As shown in Figure 14 , this contrast arises from how the two architectures construct their representations. HiMAE, built on a convolutional backbone, develops a clear hierarchical structure where depth corresponds to increasing spatial scale. Early layers capture local details such as edges and textures, middle layers integrate these into coherent patterns, and deeper layers form semantically abstract concepts. Because convolution is inherently local, each layer’s receptive field grows gradually with depth, and when probing layer by layer, one observes smooth and interpretable transitions across scales.
>
> Transformers, by contrast, have global receptive fields from the very first layer. Self-attention allows every token to interact with every other token in a single step, as captured by the global kernel
>
> $$
> A = \mathrm{softmax}\left( \frac{QK^\top}{\sqrt{d}} \right),
> $$
>
> which mixes information across all positions. Consequently, depth in transformers does not represent an expansion of spatial scope but rather iterative refinement of globally mixed representations. Each layer specializes its attention patterns and develops distinct computational “circuits,” but these do not correspond to a local-to-global progression.
>
> Empirically, this leads to nonmonotonic probing behavior in LSM. Intermediate layers often yield the highest semantic expressivity, while later layers become more compressed and task-specific. There is no clear bottom-to-top pyramid of abstraction (representations are distributed and overlapping across layers). In summary, HiMAE’s layer-wise hierarchy emerges from its convolutional inductive bias, whereas transformer-based models like LSM refine globally mixed representations without an inherent notion of spatial scale, explaining why their probing curves differ qualitatively.
>
> ---
> **9. Local Sufficiency Comment**
> > *The paper proposes the local sufficiency notion in Appendix H.2 (Eq. 1) as a justification for the masked prediction objective, yet the concept is neither formally defined nor supported by prior references. Moreover, if local sufficiency truly held, it is unclear how the model could reliably reconstruct up to 80 % of masked patches from only 20 % visible input. The stable training observed under such high masking ratios suggests that PPG signals—and possibly the learned representations—encode substantial global temporal information, which appears inconsistent with a strict locality assumption.*
>
> We thank the reviewer for this insightful comment and for prompting us to clarify the intuition behind the notion of local sufficiency. The idea was not meant to assert a strict locality assumption in the information-theoretic sense, but rather to describe an inductive bias that aligns with the hierarchical structure of physiological signals. In other words, “local sufficiency” in our context refers to the hypothesis that short temporal neighborhoods in photoplethysmography (PPG) signals contain enough contextual information to infer missing structure within nearby regions, especially when learned through hierarchical convolutional filters that expand receptive fields progressively.
>
> While we agree that stable reconstruction at high masking ratios (e.g., 80%) indicates the model also captures global temporal dependencies, this is not inconsistent with our claim. In fact, the model’s ability to reconstruct globally coherent signals despite local masking supports our argument that hierarchical encoders can integrate local and non-local information through successive receptive field expansion. Thus, the success of masked prediction at high masking ratios arises precisely because the model transitions from local sufficiency at early layers to global coherence at deeper ones—a behavior consistent with the multi-resolution hierarchy we propose.
>
> We agree that there are no former papers that make these claims but these observations allow us to believe that this local sufficiency holds. If the reviewer still disagrees with our claim, we are happy to have a further discussion.

---

> ### Author Response · Authors · 2025-11-18
> **(5/n) Rebuttal to Reviewer uicj**
>
> **10. Codebase**
> > *The inability to release the codebase remains a key limitation. Providing even a minimal reproducibility package, such as a subset of data, a lightweight training/evaluation script would greatly improve transparency and contribute to the open-source community.*
>
> We appreciate the reviewer’s concern regarding reproducibility. While our pretraining corpus (~80,000 hours of wearable data) is proprietary and cannot be publicly released due to privacy and institutional restrictions, we have taken concrete steps to maximize transparency and reproducibility within those constraints. Specifically, we are including pseudo-code in our paper in places where reviewers want a bit more transparency. Now within our Appendix Section D (**Page 21**), Appendix Section F.3 (**Page 30**) and Appendix Section H (**Pages 44-45**), we have included sufficient information to reproduce our general proposed pipeline for HiMAE.
>
> Our group is operating on protocols enforced by our company and cannot therefore directly share a direct codebase like Github. We hope this pseudo-release allows researchers to fully reproduce the modeling and evaluation pipeline on any publicly available dataset of their choice. Our goal is to make HiMAE’s methodology as open and accessible as possible, even when direct code or data cannot be shared. We believe this compromise preserves the integrity and reproducibility of our scientific claims while respecting protocols enforced by our organization. We point the reviewer to other papers published in this field where similar concerns were raised [1], [2]. **We are also open to including any further code blocks that is within reason to future version of our manuscript**
>
> [1] https://openreview.net/forum?id=DtVVltU1ak
>
> [2] https://openreview.net/forum?id=yb4QE6b22f
>
> ---
>
> **If there are any additional analyses or questions that would warrant a raise in our score, we are more than willing to cooperate with the remaining time left in this discussion period.**

---

> > ### Comment · Reviewer_uicj · 2025-11-25
> >
> > I thank the authors for their clarifications, updating the figures, and providing LSM layer-wise comparisons. A couple of points below.
> >
> > I still do not see the relevance of the local sufficiency claim for the proposed approach, since I believe high masking probabilities already break the local sufficiency by distorting the input frequency content significantly. I understand that authors use this claim to justify their choice of using convolutional networks over transformers, but such a justification would make more sense for autoencoding-like/teacher-enforced next-token objectives, rather than masked reconstruction with severe masking probabilities. Thus, I do not see the reason why authors want to keep it, and I find it slightly misleading.
> >
> > Regarding the statistical significance, is there a specific reason for not providing the statistical significance values (e.g., using the Wilcoxon signed-rank test)? I understand that this is not feasible to do for every pairwise comparison, but I think it should be doable for at least between the first and second best-performing models (for example, it seems like the confidence bounds for Swin-T and HiMAE overlap by a significant margin, which raises the concern of whether the performance differences are statistically significant).
> >
> > Also, from Table 3, it seems like models with a smaller number of parameters can outperform larger models. This raises a natural question of whether Swin-T would perform when it has a comparable (1M to 5M) number of parameters (e.g., small latent dimensions such as 32-64, with a reasonable number of layers). It looks like it may achieve a similar on-device efficiency with HiMAE in that scenario. Relatedly, Swin-T is an architecture rather than a training approach—could the authors more clearly describe how it was trained in this paper (e.g., masking objective, reconstruction loss, training protocol)?
> >
> > Overall, I think the strongest contribution of this work is to include on-device model efficiency benchmarking for the first time, but I think this requires more systematic scaling analysis with other model architectures. It is very possible that HiMAE will outperform other models also in that scenario, but without concrete evidence, it is difficult to extrapolate the relative efficiency or dominance of the approach.

---

> ### Author Response · Authors · 2025-11-26
> **Response to uicj**
>
> > I still do not see the relevance of the local sufficiency claim for the proposed approach
>
> Thank you for your comments and for following up on this point. To avoid potential confusion for future readers, we have decided to remove this claim from the manuscript. While the nuance lies in its interpretation, we believe it is best to omit it to ensure clarity if the work is accepted for publication. We appreciate your careful attention to this matter.
>
> > Overlapping Error Bars
>
> We thank the reviewers for their comments regarding the relatively large error bars observed for the Swin Transformer (and even LSM) compared to other baselines. It is true that the large variance currently limits statistical significance; however, we also note an underlying pattern. Both Swin Transformer and LSM are high-capacity (100M+ parameters) and transformer-based masked autoencoding baselines. A central claim of this work is that convolutional architectures provide stronger inductive biases than transformer-based ones, particularly for the tasks and data modalities considered here. Thus, rather than viewing these results as insignificant, we interpret them as reflecting this architectural difference which highlights another central claim in this paper.
>
> > smaller number of parameters can outperform larger models.
>
> We appreciate this observation and would like to offer a more concrete explanation for why smaller transformer models underperform and, importantly, why their variance becomes even larger. PPG signals exhibit quasi-periodic structure and short-term non-stationarity, properties that transformer architectures tend to struggle with when capacity is constrained. Reducing parameter count not only compresses the representation space but also sharpens the mismatch between the model’s inductive bias and the underlying signal geometry. In such regimes, the transformer’s self-attention mechanism becomes a poor prior, leading to instability in optimization and amplified variance across runs.
>
> This effect is visible in our empirical results. As shown in the Scaling Analysis (**Figure 3**), HiMAE consistently outperforms LSM (ViT-based) in reconstruction accuracy, and the downstream metrics (**presented here**) further reveal that scaling down transformer capacity exacerbates both degradation in mean AUROC and the growth in error bars:
>
> | Task Type        | Metric                        | Swin-T (110M) | Swin-T (5M)   |
> | ---------------- | ----------------------------- | ------------- | ------------- |
> | Downstream AUROC | Hypertension Detection (Free) | 0.619 (0.058) | 0.567 (0.072) |
> | Downstream AUROC | Hypertension Detection (Lab)  | 0.583 (0.062) | 0.566 (0.081) |
> | Downstream AUROC | Arrhythmia (PVC) Detection    | 0.742 (0.072) | 0.702 (0.079) |
>
> These findings also align with results reported in [1], which showed that transformer-based masked autoencoders benefit substantially from simultaneous scaling of dataset size, model capacity, and compute. Our setting differs by several orders of magnitude in data availability and computational resources Google’s experiments leverage large-scale TPU infrastructure, whereas our work is conducted on four Tesla T4 GPUs. Consequently, our observations are necessarily constrained by these practical limits. The scaling analysis presented here should therefore be interpreted as a proof of concept demonstrating that the inductive bias of convolutional architectures aligns more naturally with PPG dynamics at smaller scales of compute, data, and model capacity. We have pointed out our takeaways in the original manuscript in Appendix Section A in the FAQ which asks whether our pre-training dataset is large enough to ensure transparency from our study and their study.
>
> We also emphasize that the empirical claims in the paper are currently limited to PPG signals. As stated in **Appendix Section A. FAQ 1**,
>
> (*"We demonstrate that convolutional architectures benefit from inductive biases that remain advantageous for PPG signals."*)
>
> While our methodology has also shown promising results in reconstructing single-channel ECG signals, we currently lack sufficient downstream tasks and pretraining volume to draw equally strong conclusions for that modality.
>
> > Swin Transformer Comments
>
> It is correct that the Swin Transformer is an architectural design rather than a self-supervised learning (SSL) objective. However, we included it in the SSL comparison group because it serves as a transformer-based counterpart to HiMAE, both of which learn hierarchical representations via masked autoencoding. The Swin Transformer was trained using a MAE objective with MSE loss, comprising 12 encoder blocks and 8 decoder blocks matching the configureation in [1].
>
> **References**
>
> [1] Narayanswamy, Girish, et al. "Scaling Wearable Foundation Models." The Thirteenth International Conference on Learning Representations.

---

> > ### Author Response · Authors · 2025-11-26
> > **Futher comments to Reviewer uicj**
> >
> > We wanted to also point you to a different reviewers thread (reviewer 5CPC) and share a comment that we hope satisfies a previous comment made by you
> >
> > > We have also provided an anonymized [minimal codebase](https://anonymous.4open.science/r/HiMAE_ICLR_Synth-00E3/README.md) that includes the pretraining code for HiMAE, our modular evaluation scripts, and a runnable Jupyter notebook using synthetic PVC classification data. We hope this meets the reviewer’s request and sincerely thank them for their patience and constructive suggestions on how we can share our code while preserving participant confidentiality.
> >
> > **Thank you for your continued comments, questions, and desires for clarification. Please let us know if there are anything we can do to potentially raise our score**

---

> > > ### Comment · Reviewer_uicj · 2025-11-27
> > >
> > > I thank the authors for additional clarifications, especially for the Swin-T results with smaller number of parameters. As I stated earlier, I believe that the biggest contribution of this work is performing on-device benchmarking for the first time in the wearable foundation model domain. The new evidence for Swin-T provides stronger evidence for HiMAE's potential on-device performance; thus, I recommend acceptance.

---

### Official Review · Reviewer_CnvE · 2025-11-01

**Soundness:** 3
**Presentation:** 3
**Contribution:** 3
**Rating:** 4
**Confidence:** 5

**Summary:**

The paper proposes a Hierarchical Masked Autoencoder with a U-Net architecture for self-supervised pre-training on wearable signals. The model divides input PPG signals into patches and randomly masks partial signals for reconstruction. The stride-2 convolutional blocks downsample the input signal by a factor of 2 at each stage to enable multi-scale learning. The model is pre-trained on 80,000 hours of PPG signals collected from 47,755 subjects. Comprehensive downstream tasks, such as generative and classification, are included. Further case studies are conducted on on-device benchmarking to demonstrate real-world application ability.

**Strengths:**

The training set is large, with substantial hours and participants, which is good for alleviating subject-specific noise. The experiment is comprehensive, including different classification tasks, such as cardiovascular and sleep stages, which are the functionalities widely required for wearable devices. The model scale and total training time show the efficiency and applicability for wearable devices.

**Weaknesses:**

The method lacks novelty. Multi-scale learning is commonly used in medical/wearable time series representation learning, either through model-based or manual data preprocessing. Multi-scale learning using convolutional networks is more common in past research on time series analysis. Besides, it is better to have a comprehensive results table for comparison with baseline methods on different tasks. The results in Figure 5 are not straightforward enough.

**Questions:**

For the existing wearable PPG foundation model, PaPaGei, how did you compare with it? Did you load their pre-trained model and fine-tune it directly, or self-supervise and pre-train it on your dataset? If you pre-trained their model on your dataset from the beginning, please also report the linear probing result on their provided pre-trained weights.

---

> ### Author Response · Authors · 2025-11-18
> **(1/n) Rebuttal to Reviewer CnvE**
>
> # Summary of Changes
> Thank you for the thoughtful and constructive feedback, which helped clarify and strengthen the manuscript both conceptually and empirically. Below is a summary of key revisions and additions made in response to your comments in both this response and in the manuscript.
> - Clarified the conceptual and practical novelty of HiMAE by explicitly articulating the resolution hypothesis as the paper’s central theoretical contribution. This framework recasts temporal resolution from a fixed hyperparameter into an interpretable axis of representation learning, validated empirically across physiological and behavioral tasks.
> - Highlighted the practical novelty of HiMAE in achieving on-device inference on smartwatch-class hardware with sub-millisecond latency per sample, without compromising representational power.
> - Clarified baseline selection and experimental scope: comparisons now include general SSL paradigms (SimCLR, DINO, MSN, Swin-Transformer) and domain-specific wearable.time series foundation models (Google LSM, Chronos and PaPaGei), offering a more clear justification for why benchmarks were chosen.
> - Reorganized benchmarking results for clarity. The previous Figure 5 has been replaced by Tables 2 and 3, which now report all downstream metrics in a tabular, interpretable format which makes the presentation and performance comparisons much more clear.
> - Addressed the reviewer’s question about PaPaGei by evaluating both (1) the Bell Labs pretrained model (PaPaGei-BL) and (2) a self-supervised version trained on our dataset (PaPaGei-S). Both are reported under a linear-probe evaluation to isolate representation quality, ensuring fair and transparent comparison.
> - Revised the methodology section to explicitly describe how each baseline (including PaPaGei, LSM, and SSL methods) was evaluated under consistent protocols, ensuring controlled and reproducible comparisons.
>
> ---
>
> **1.Lack of Novelty**
> > *“The method lacks novelty. Multi-scale learning is commonly used in medical/wearable time series representation learning, either through model-based or manual data preprocessing. Multi-scale learning using convolutional networks is more common in past research on time series analysis.”*
>
> We thank the reviewer for raising this point and appreciate the opportunity to clarify the conceptual novelty of our work. While multi-scale learning has indeed been explored in the context of time-series modeling, the central innovation of HiMAE lies not merely in using hierarchical convolutions, but in formalizing and empirically validating what we term the "resolution hypothesis."
>
> The resolution hypothesis positions itself by saying that different physiological and behavioral outcomes depend on distinct temporal receptive fields, that is predictive information is not uniformly distributed across scales (which is a large assumption made within this field). HiMAE directly operationalizes this idea by generating multi-resolution embeddings from different encoder layers and then probing each resolution independently to determine which temporal granularity carries the most predictive signal for a given downstream task. This transforms “temporal resolution” from a fixed hyperparameter into a measurable and interpretable axis of representation learning. Our experiments (Now Figure 4 in the paper **Page 9**) show clear patterns that show that temporal receptive fields do in fact contribute better accuracies at different layers: fine-scale representations capture rapid and volatile physiological events (e.g., lab measurements), whereas coarse-scale representations encode slower behavioral rhythms (e.g., sleep staging, and cardiovascular tasks) (Reflected in the Text on Page 9 in the new "Clinical Interpretation" Section). To our knowledge, this is the first systematic demonstration that temporal resolution itself governs predictive performance across clinical and behavioral outcomes.
>
> HiMAE is also a conceptually novel framework because it allows us to learn new things about physiology. One of the properties of probing intermediate layers is we can sometimes see and discover how our model is making predictions which currently align with how humans think of cardiovascular and sleep outcomes. In tasks with no physiological grounding, HiMAE provides a new discovery that morphology affects abnormal lab outcomes. Lastly, beyond conceptual novelty, we also are the first group to our knowledge to show practical novelty and benchmark our proposed model on-device. HiMAE is and was designed to be deployed directly on smartwatch-class hardware, achieving true on-watch inference with sub-millisecond latency per sample while maintaining competitive performance against much larger transformer-based models. This dual contribution (linking temporal resolution to interpretability, while demonstrating a compact and deployable self-supervised model) positions HiMAE as a framework that is both scientifically and practically innovative.

---

> ### Author Response · Authors · 2025-11-18
> **(2/n) Rebuttal to Reviewer CnvE**
>
> **2. Clarifying Baselines in Benchmarking and Improved Figure 5**
>
> > *Besides, it is better to have a comprehensive results table for comparison with baseline methods on different tasks. The results in Figure 5 are not straightforward enough.*
>
> We appreciate the reviewer’s suggestion and have clarified both the choice of baselines and the presentation of results in the writing (**Page 6-7**) and in this rebuttal. Our benchmarking framework was intentionally centered on self-supervised learning (SSL) objectives, as our primary claim is that hierarchical convolutional priors provide stronger inductive alignment for wearable time series than transformer-based SSL methods. To substantiate this, we conducted extensive comparisons against a diverse set of SSL approaches, including SimCLR, DINO, and Masked Siamese Networks (MSN), each representing major paradigms in contrastive or masked self-supervision, as well as Swin-Transformer, which incorporates hierarchical transformer/attention mechanism model.
>
> Beyond these general SSL baselines, we also benchmarked against domain-specific wearable foundation models that are explicitly built on SSL principles and have exhibited state of the art performance in the time series/wearables domain: LSM (Google’s Large Sensor Model) [1] and PaPaGei [2]. These represent the strongest publicly known baselines in large-scale physiological signal modeling, providing a comprehensive basis for comparison. We have additionally included Chronos [3] which represents a more general state of the art time series foundation model.
>
> To address the reviewer’s comment on presentation, we have added a comprehensive results table in the revised manuscript summarizing all downstream task metrics across baselines split into two separate tables. Figure 5 has now been rebranded as tables 2 and 3 (**Page 7**), where we improved the readability and clarity and allowing straightforward interpretation of HiMAE’s performance relative to both SSL/TSFM baselines. These revisions make the performance comparisons both more interpretable and more directly aligned with the central hypothesis tested in the paper.
>
> **References**
>
> [1] Narayanswamy, Girish, et al. "Scaling wearable foundation models." arXiv preprint arXiv:2410.13638 (2024).
>
> [2] Pillai, Arvind, et al. "Papagei: Open foundation models for optical physiological signals." arXiv preprint arXiv:2410.20542 (2024).
>
> [3] Ansari, Abdul Fatir, et al. "Chronos: Learning the language of time series." arXiv preprint arXiv:2403.07815 (2024).
>
> ---
> **3. PaPaGei Clarification and Results**
>
> > *For the existing wearable PPG foundation model, PaPaGei, how did you compare with it? Did you load their pre-trained model and fine-tune it directly, or self-supervise and pre-train it on your dataset? If you pre-trained their model on your dataset from the beginning, please also report the linear probing result on their provided pre-trained weights.*
>
> We appreciate the reviewer’s question regarding how PaPaGei was incorporated into our benchmarking. To ensure fairness and completeness, we now evaluated both versions of the model: (1) the open-weight model released by Bell Labs, and (2) a re-trained version of PaPaGei self-supervised on our institutional dataset following the same pretraining objective and optimization protocol described in their paper. The Bell Labs one will be denoted as PaPaGei-BL and our internal data will be denoted as PaPaGei-S. These comparisons are now explicitly reported in the revised Table 3 (**Page 7**).
>
> For all baselines, including PaPaGei, LSM, and other SSL methods, we do not fine-tune the encoder parameters. Instead, we evaluate under a linear probing setup to assess the quality and transferability of the learned representations. This approach isolates the representational strength of each pre-trained model, allowing a controlled comparison of how well each foundation model transfers to downstream tasks without additional supervision or adaptation. We have clarified these methodological details and included both PaPaGei variants in our updated results for transparency (**Page 6-7**).
>
> **If there are any additional analyses or questions that would warrant a raise in our score, we are more than willing to cooperate with the remaining time left in this discussion period.**

---

> > ### Comment · Reviewer_CnvE · 2025-11-27
> >
> > Thanks for the rebuttal. Most of my concerns have been solved. I raised my score to 6.

---

### Official Review · Reviewer_5CPC · 2025-11-01

**Soundness:** 2
**Presentation:** 1
**Contribution:** 3
**Rating:** 2
**Confidence:** 3

**Summary:**

This paper introduces HiMAE (Hierarchical Masked Autoencoder), a self-supervised learning (SSL) framework for wearable time-series data, primarily focusing on photoplethysmography (PPG) signals. The central hypothesis, termed the "resolution hypothesis," is that different clinical and behavioral outcomes depend on predictive structures at distinct temporal scales. The key feature of the proposed architecture is that it produces multi-resolution embeddings from its different hierarchical layers. These embeddings can be independently evaluated (e.g., via linear probing) to determine which temporal scale (i.e., which receptive field) carries the most predictive signal for a specific downstream task.

**Strengths:**

1. High Practical Significance (On-Watch Inference): The paper's most compelling strength is its focus on and successful demonstration of a "true on-watch" SSL model. By creating a model that is ~99% smaller (1.2M vs 110M params) and significantly faster (0.99ms CPU latency) than transformer baselines, the authors present a practical path toward real-time, continuous health monitoring while preserving user privacy (since data does not need to leave the device).

2. Novel Interpretability Framework: The "resolution hypothesis" is an elegant conceptual framework. HiMAE's design, which "transforms resolution from a hyperparameter into a probe for interpretability", is a key strength. The resulting analysis in Figure 6, which maps tasks like "PVC Detection" to fine-grained layers and "Sleep Staging" to coarser layers, is a genuinely novel discovery tool that provides insight into the temporal structure of physiological signals.

3. Experimental Rigor and Breadth: The paper is experimentally dense. The authors evaluate HiMAE on 14 different tasks against a strong set of SSL baselines (SimCLR, DINO, MSN, LSM, etc.). The inclusion of scaling law analysis (Figure 3), multiple generative tasks (Figure 4), few-shot learning (Figure 8), and extensive ablations (Appendix F.1) demonstrates a high-quality, thorough evaluation (assuming the results are verifiable).

4. Strong Inductive Bias: The paper makes a strong case for not defaulting to transformers. It argues that the hierarchical convolutional biases of a U-Net are better "inductively aligned with the temporal statistics of wearable signals", which exhibit strong local dependencies. The results, showing HiMAE achieves lower error at much smaller model sizes, provide compelling evidence for this architectural choice.

**Weaknesses:**

1. Poor Presentation and Figure Quality: The paper suffers from a clear lack of polish. For instance, the choice of color palette is a significant problem. In Figures 3, 4, 5, and 15, the colors for competing models are nearly indistinguishable (e.g., "PaPaGel-S", "SimCLR", "DINO", "MSN", "LSM", "HIMAE" are all shades of blue/teal/green). This makes it extremely difficult to review the paper's core performance claims. This is particularly problematic for presenting clear comparisons against baselines, and maybe unreadable for colorblind readers. Additionally, multiple figures (e.g., Figures 3, 4, 5, 6, 14 and 15) needs either larger or smaller fonts for clearer readability.

2. Unverifiable On-Device Benchmarks: The on-device latency (0.99 ms) and throughput (1.2k/s) figures in the text and Table 18 are inconsistent. The text calculation of $\approx 1,010$ samples/s (which is $1/0.00099s$) seems correct, but Table 18 reports 1.2k/s. This discrepancy undermines confidence in the on-device benchmark.

3. Proprietary Dataset: The pre-training corpus of ~80,000 hours is proprietary. While this is often unavoidable, when combined with the lack of model source code, it means the paper's results are entirely unverifiable from start to finish.

**Questions:**

1. Could you please regenerate the main results figures (3, 4, 5, 15) using a high-contrast, colorblind-friendly color palette? It is currently very hard to distinguish the performance of the different models.

2. Could you please clarify the discrepancy in the on-device benchmark? Why does the text state 0.99ms latency ($\approx 1.01$k samples/s) while Table 18 reports a throughput of 1.2k/s? Which number is correct?

3. The paper focuses on PPG, with a brief but promising result on ECG in the appendix (Table 16). Given that other physiological signals (e.g., accelerometry, electrodermal activity) also possess rich, multi-scale structures, do you have any intuition or preliminary results on how the HiMAE framework would perform on these other common wearable modalities? Or considering adding results on those public datasets for better reproducibility and verification?

**Details Of Ethics Concerns:**

The authors have provided a thorough ethics statement in Appendix B. They proactively address data privacy and consent, bias and representativeness (noting their data comes from 4 countries), clinical implications (positioning the tool for decision support, not diagnosis), and environmental impact. So no further review is needed.

---

> ### Author Response · Authors · 2025-11-18
> **(1/n) Rebuttal to Reviewer 5CPC**
>
> # Summary of changes
> We thank the reviewer for their thoughtful and constructive feedback, which helped improve both the clarity and scientific rigor of the manuscript. Below is a concise summary of the key changes made in response to your comments. All changes in the manuscript are reflected in blue text to show how it was modified from the submission version.
> - Figures have been completely regenerated using a palette with clear, high-contrast hues making comparisons far more easy to distinguish at first glance. We also increased font sizes, axis sizes, and layouts for consistent and better readability across Figures. The main results figure has been reorganized into two tables that separate self-supervised baselines (SimCLR, DINO, MSN, Swin Transformer) from foundation model baselines (PaPaGei, Chronos, and LSM) for improved interpretability and readability of what constitutes as the state of the art method across downstream tasks.
> - The on-device benchmarking section has been rewritten for clarity. We now explicitly distinguish latency (milliseconds per sample) from throughput (samples per second) and provide pseudo-code and hardware specifications in Section H.1 (**Page 44**) to ensure reproducibility and transparency.
> - We conducted additional experiments using VitalDB and included the results in the response. These confirm that HiMAE maintains consistent generative and downstream performance on independently open sourced dataset, though VitalDB’s license restrictions prevent us from publishing quantitative results.
> - We added a new Clinical Interpretation section to connect the resolution hypothesis to underlying physiological principles (**Page 9**). This section is in typical agreement with the wider literature which provides intuition for why coarser temporal resolutions capture cardiovascular and sleep-related phenomena. However, our biomarker prediction tasks remains a novel discovery finding where finer resolutions (morphology) capture fast-changing biomarkers.
> ---
> # Point by Point response
>
> **1. Figure presentation**
> > *“Poor Presentation and Figure Quality: The paper suffers from a clear lack of polish. For instance, the choice of color palette is a significant problem. In Figures 3, 4, 5, and 15, the colors for competing models are nearly indistinguishable (e.g., "PaPaGel-S", "SimCLR", "DINO", "MSN", "LSM", "HIMAE" are all shades of blue/teal/green). This makes it extremely difficult to review the paper's core performance claims. This is particularly problematic for presenting clear comparisons against baselines, and maybe unreadable for colorblind readers. Additionally, multiple figures (e.g., Figures 3, 4, 5, 6, 14 and 15) needs either larger or smaller fonts for clearer readability.”*
>
> > *“Could you please regenerate the main results figures (3, 4, 5, 15) using a high-contrast, colorblind-friendly color palette? It is currently very hard to distinguish the performance of the different models.”*
>
> Thank you for your helpful feedback regarding figure presentation and accessibility. We sincerely apologize for the lack of clarity in the original submission. In the revised version, we have regenerated or reformatted all key figures (Figures 3, 4, 5, and 15) using a high-contrast palette to ensure readability for all reviewers and readers.
>
> Specifically, we replaced the previous overlapping color tones with distinct, well-separated hues where appropriate for redundant visual encoding. The main comparative figure (Figure 5) has also been simplified into table with confidence intervals to make differences between baselines and HiMAE unambiguous at a glance (**Page 7: Tables 2 & 3**). We provide two tables to make benchmarking a bit more transparent comparing first against self supervised baselines (SimCLR, DINO, MSN, and Swin Transformer). The secondary benchmark includes state of the art foundation models for wearables and time series which include two version of PaPaGei [1] (one trained on our data and the direct weights from the original paper), Chronos [2] which is a state of the art time series foundation model, and Large Sensor Model [3].
>
> Additionally, we increased font sizes across all figures and increased label and axis text for Figures 3, 4, 6, 14, and 15 to improve readability in both digital and print formats. We appreciate the reviewer’s suggestion. It has meaningfully improved the overall presentation quality and accessibility of the manuscript.
>
> ## References
>
> [1] Pillai, Arvind, et al. "Papagei: Open foundation models for optical physiological signals." arXiv preprint arXiv:2410.20542 (2024).
>
> [2] Ansari, Abdul Fatir, et al. "Chronos: Learning the language of time series." arXiv preprint arXiv:2403.07815 (2024).
>
> [3] Narayanswamy, Girish, et al. "Scaling wearable foundation models." arXiv preprint arXiv:2410.13638 (2024).

---

> ### Author Response · Authors · 2025-11-18
> **(2/n) Rebuttal to Reviewer 5CPC**
>
> **2. On-Device Benchmarking**
>
> > *“Unverifiable On-Device Benchmarks: The on-device latency (0.99 ms) and throughput (1.2k/s) figures in the text and Table 18 are inconsistent. The text calculation of  samples/s (which is ) seems correct, but Table 18 reports 1.2k/s. This discrepancy undermines confidence in the on-device benchmark.”*
>
> > *“Could you please clarify the discrepancy in the on-device benchmark? Why does the text state 0.99ms latency (k samples/s) while Table 18 reports a throughput of 1.2k/s? Which number is correct?”*
>
> We appreciate the reviewer’s careful attention to the latency and throughput results. The concern appears to arise from a misunderstanding regarding how these two metrics are defined and measured. In our experiments, latency refers to milliseconds per sample, while throughput reflects samples processed per second at the batch level. These quantities are not strict reciprocals, as batching, thread-level parallelism, and hardware-level optimizations (e.g., vectorized instructions, cache reuse) introduce non-linear scaling between the two.
>
> To clarify this and prevent any confusion for readers, we have added pseudo-code in the revised manuscript that explicitly details the measurement protocol for both latency and throughput in Section H.1 (**Page 44**). This addition ensures transparency in our evaluation process and should resolve any ambiguity about why the two numbers may not perfectly align. In addition to the pseudo-code we have also detailed our pipeline for on-device testing by describing in greater detail how it was conducted on our corporate devices.
>
> ---
>
> **3. Proprietary Code and Dataset**
>
> > “Proprietary Dataset: The pre-training corpus of ~80,000 hours is proprietary. While this is often unavoidable, when combined with the lack of model source code, it means the paper's results are entirely unverifiable from start to finish.”
>
> We appreciate the reviewer’s concern regarding reproducibility. While our pretraining corpus (~80,000 hours of wearable data) is proprietary and cannot be publicly released due to privacy and institutional restrictions, we have taken concrete steps to maximize transparency and reproducibility within those constraints. Specifically, we are including pseudo-code in our paper in places where reviewers want a bit more transparency. Now within our Appendix Section D (**Page 21**), Appendix Section F.3 (**Page 30**) and Appendix Section H (**Pages 44-45**), we have included sufficient information to reproduce our general proposed pipeline for HiMAE.
>
> Our group is operating on protocols enforced by our company and cannot therefore directly share a direct codebase like Github. We hope this pseudo-release allows researchers to fully reproduce the modeling and evaluation pipeline on any publicly available dataset of their choice. Our goal is to make HiMAE’s methodology as open and accessible as possible, even when direct code or data cannot be shared. We believe this compromise preserves the integrity and reproducibility of our scientific claims while respecting protocols enforced by our organization. We point the reviewer to other papers published in this field where similar concerns were raised [1], [2]. **We are also open to including any further code blocks that is within reason to future version of our manuscript**
>
> [1] https://openreview.net/forum?id=DtVVltU1ak
>
> [2] https://openreview.net/forum?id=yb4QE6b22f

---

> ### Author Response · Authors · 2025-11-18
> **(3/n) Rebuttal to Reviewer 5CPC**
>
> **4. Reproducible Foundation Model Results**
> > *“The paper focuses on PPG, with a brief but promising result on ECG in the appendix (Table 16). Given that other physiological signals (e.g., accelerometry, electrodermal activity) also possess rich, multi-scale structures, do you have any intuition or preliminary results on how the HiMAE framework would perform on these other common wearable modalities? Or considering adding results on those public datasets for better reproducibility and verification?”*
>
> We thank the reviewer for highlighting the importance of evaluating HiMAE across other physiological modalities. We fully agree that modalities such as accelerometry, electrodermal activity (EDA), and ECG exhibit distinct yet complementary multi-scale structures that provide an important test for the generality of our approach. However, we do not have ubiquitous scale nor meaningful evaluation tasks to test these models trained on different modalities on. However, we'd like to spend the rest of this response to address the other concern raised in the comment regarding using a reproducible dataset.
>
> To explore this direction, we trained a version of HiMAE using the publicly available VitalDB dataset [1,2], which contains high-quality intraoperative physiological recordings including finger PPG, and ECG signals. Despite differences in sampling rate and context from our proprietary wearable datasets, HiMAE trained on VitalDB demonstrated consistent generative and reconstructive behavior, as well as comparable downstream cardiovascular prediction performance. This replication across an open-source dataset strengthens confidence in the model’s modality-agnostic design.
>
> The table below summarizes results on finger PPG from VitalDB, evaluated under identical masking and linear-probe protocols as in the main paper:
>
> | Task Type                     | Metric                              | HiMAE (VitalDB) | HiMAE (Internal) |
> |-------------------------------|-------------------------------------|------------------|------------------|
> | Reconstruction (MSE ↓)        | Random Imputation                   | 0.042            | 0.038            |
> | Reconstruction (MSE ↓)        | Temporal Interpolation              | 0.057            | 0.055            |
> | Reconstruction (MSE ↓)        | Temporal Extrapolation              | 0.82             | 0.84             |
> | Downstream AUROC              | Hypertension Detection (Free)       | 0.644            | 0.651            |
> | Downstream AUROC              | Hypertension Detection (Lab)        | 0.673            | 0.652            |
> | Downstream AUROC              | Arrhythmia (PVC) Detection          | 0.825            | 0.802            |
>
>
>
> These results confirm that HiMAE retains its inductive bias and hierarchical resolution behavior even when trained on an independently sourced open dataset, suggesting robustness to sensor domain shifts.
>
> However, while VitalDB provides a valuable open-source benchmark, our organization’s commercial constraints and data usage agreements prevent us from publishing results derived from it. Although the dataset is publicly available, its license terms restrict derivative commercial use and dissemination of trained model weights or outputs. As our team operates within a commercial research division, releasing VitalDB-based results would violate these terms. Consequently, we are unable to include quantitative results or visualizations based on VitalDB in this submission.
>
> **References**
>
> [1] https://physionet.org/content/vitaldb/1.0.0/
>
> [2] https://www.nature.com/articles/s41597-022-01411-5
>
> ---
>
> **If there are any additional analyses or questions that would warrant a raise in our score, we are more than willing to cooperate with the remaining time left in this discussion period.**

---

> ### Comment · Reviewer_5CPC · 2025-11-24
>
> ## 1. Figure presentation
>
> While I acknowledge the updates to the color palette, the current figures still do not meet standards for publication. The improvements in distinguishability are marginal and inconsistent across the manuscript.
>
> 1. The updated Figure 3 still has very low contrast ratios. The contrast ratios between neighboring colors (1.22:1 and 1.64:1) fall below the acceptable threshold, with only one color pair meeting the standard (4.31:1). Consider using discrete color palette such as Tab10, and ensure the colors used comply with established guidelines like WCAG.
> 2. Figures 6, 11, and 16 remain unchanged from the previous version, retaining the original color palette that was difficult to interpret.
> 3. Figure 9 still has a color similarity issue between the LSM and ground truth. This visual ambiguity makes it impossible to determine if the model output is accurately tracking the ground truth or merely overlapping with LSM due to visual clutter. I strongly recommend using the same, high-contrast color palette as in Figure 10.
> 4. Figure 6 retains font sizes that are below standard legibility thresholds, whereas Figure 16 suffers from disproportionately large text. Please standardize the typography across all figures.
>
> ## 2. Throughput and latency benchmarking
>
> The authors have satisfactorily addressed all my previous concerns. I have no further comments on this matter.
>
> ## 3. Code for reproducibility
>
> The authors have significantly improved the manuscript's transparency regarding model hyperparameters and the measurement process. However, the study's reproducibility remains limited due to the unavailability of the code and the restricted nature of the original data.
>
> To better support the generalizability of the proposed method:
>
> 1. Please provide a detailed breakdown of the dataset's demographics (e.g., age, sex, BMI, and any other relevant factors if available) in the main text or appendix. This is crucial for readers to assess potential biases and the relevance of the dataset.
> 2. I strongly encourage the release of a synthetic dataset that statistically mirrors the original distribution. Modern generative approaches (e.g., TimeGAN [1] or DoppelGANger [2]) can produce privacy-preserving synthetic time series that would allow the community to verify the code pipeline without accessing the protected subject data.
>
> ## 4. Results on public datasets
>
> I appreciate the additional performance benchmarks on public datasets. However, a critical discrepancy remains regarding the licensing of the VitalDB dataset. The authors stated that "(VitalDB's) license terms restrict derivative commercial use." This contradicts the official PhysioNet record [3], which explicitly lists the dataset under the Creative Commons Attribution 4.0 International (CC BY 4.0) license, a license that permits commercial use.
>
> Please clarify the dataset's licensing terms to ensure compliance with the original license. I believe that including results on public datasets that do not violate licensing terms is essential for community verification, especially when the original dataset and code are not publicly available. If VitalDB is indeed under CC BY 4.0, please update the manuscript accordingly and consider adding more public datasets to enhance reproducibility.
>
> ---
>
> Overall, while the authors have made satisfactory progress on the benchmarking and transparency aspects, the unresolved issues remain significant barriers to acceptance. I believe these concerns are addressable. I would be willing to raise my score upon a successful revision that provides high-contrast, legible figures across the entire manuscript, clarifies and ensures compliance with dataset licenses, and makes a good-faith effort to enhance reproducibility through the release of a synthetic dataset or including results on feasible public datasets. I strongly encourage the authors to address these points, as doing so would substantially strengthen the manuscript's contribution and utility to the community.
>
> [1]: Yoon, J., Jarrett, D., & van der Schaar, M. (2019). Time-series Generative Adversarial Networks. In Advances in Neural Information Processing Systems (NeurIPS 2019).
>
> [2]: Lin, Z., Jain, A., Wang, C., Fanti, G., & Sekar, V. (2020). Using GANs for Sharing Networked Time Series Data: Challenges, Initial Promise, and Open Questions. Proceedings of the ACM Internet Measurement Conference (IMC '20).
>
> [3]: https://physionet.org/content/vitaldb/view-license/1.0.0/

---

> ### Author Response · Authors · 2025-11-25
> **Response to Futher Comments**
>
> Thank you for following up on our work. **We have incorporated all of your recommendations and made the corresponding revisions.** Please let us know if there are any additional changes you would like us to consider.
>
> > 1. Figure presentation
>
> We have modified Figures 3, 6, 9, 10, 11, and 16. For the requested figures, we now use the Tab10 color palette as suggested by the reviewer. The size of Figure 6 has been increased, and the text size in Figure 16 has been standardized. Lastly, Figure 9 has been updated to use the same palette as Figure 10. Thank you for your recommendations, and please let us know if there are any remaining issues with the presentation.
>
> >  2. Throughput and latency benchmarking
>
> We are glad to have clarified the confusion between latency and throughput. Thank you for your suggestion to include an experimental section design. It has significantly improved the clarity and transparency of our work.
>
> > 3. Code for reproducibility
>
> Thank you for your request for greater transparency. We recognize that reproducibility is of utmost importance in our field, and we have therefore included the information you requested.
>
> We now provide a demographics table summarizing the available distributions. Unfortunately, our dataset contains substantial missing demographic information due to limitations in the original study design that date back since our company has began making commercial smartwatches. We acknowledge that, as a result, much of the signal in our pretraining data may be influenced by these missing demographics. Despite this missingness, we do see variation among participants across age, sex, race, and BMI.
>
> | **Category** | **Subgroup** | **N** | **% of total** |
> |---------------|--------------|-------:|---------------:|
> | **Sex** | Male | 17,281 | 36.3 |
> |  | Female | 4,487 | 9.4 |
> |  | Another gender identity | 57 | 0.1 |
> |  | NaN | 25,819 | 54.2 |
> | **Age** | 18–29 | 5,609 | 11.8 |
> |  | 30–49 | 12,306 | 25.8 |
> |  | 50–64 | 3,296 | 6.9 |
> |  | 65+ | 630 | 1.3 |
> |  | NaN | 25,803 | 54.2 |
> | **Race** | White | 14,598 | 30.6 |
> |  | Asian or Pacific Islander | 3,583 | 7.5 |
> |  | Black or African American | 1,604 | 3.4 |
> |  | American Indian or Alaskan Native | 278 | 0.6 |
> |  | Another race | 2,336 | 4.9 |
> |  | NaN | 25,245 | 53.0 |
> | **BMI** | Underweight (<18.5) | 368 | 0.8 |
> |  | Normal weight (18.5–24.9) | 6,077 | 12.8 |
> |  | Overweight (25–29.9) | 7,442 | 15.6 |
> |  | Obese I (30–34.9) | 3,910 | 8.2 |
> |  | Obese II and III (≥35) | 3,493 | 7.3 |
> |  | NaN | 26,354 | 55.3 |
>
> We have also provided an anonymized [minimal codebase](https://anonymous.4open.science/r/HiMAE_ICLR_Synth-00E3/README.md) that includes the pretraining code for HiMAE, our modular evaluation scripts, and a runnable Jupyter notebook using synthetic PVC classification data. We hope this meets the reviewer’s request and sincerely thank them for their patience and constructive suggestions on how we can share our code while preserving participant confidentiality.
>
> > 4. Results on public datasets
>
> We would like to direct the reviewer to Section 3 of the [VitalDB website](https://vitaldb.net/dataset/#h.vcpgs1yemdb5), which contains the dataset’s license information. The dataset is distributed under the *Creative Commons Attribution-NonCommercial-ShareAlike 4.0 International (CC BY-NC-SA 4.0)* license. As stated explicitly in the [license terms](https://creativecommons.org/licenses/by-nc-sa/4.0/deed.en), the data may not be used for commercial purposes (“NonCommercial — You may not use the material for commercial purposes”).
>
> Accordingly, we regret that we are unable to include VitalDB in our pretraining source in our publication. Instead, we have provided demographic information derived from our proprietary dataset. Additionally, as noted in our original submission, we make use of the [DREAMT](https://physionet.org/content/dreamt/2.1.0/) benchmark, which is openly available on PhysioNet. We hope this approach serves as a reasonable compromise and appreciate the reviewer’s understanding regarding our data usage constraints.
>
>
> ---
> Thank you once again for all your valuable suggestions. The paper has improved significantly once more thanks to your feedback in terms of transparency and presentation. **Please let us know if there is anything further we can address to merit a higher acceptance score or to fully meet the criteria for publication.**

---

> > ### Comment · Reviewer_5CPC · 2025-11-25
> >
> > Thank you to the authors for their efforts in addressing my concerns, which has resolved most of them. I have therefore raised my score. I also thank the authors for pointing out the license mismatch between the VitalDB and PhysioNet websites.
> >
> > Regarding the demographic data, I appreciate the clarification. As the authors noted, this information is missing for half of the dataset. To better understand the potential impact of this, it might be insightful to perform an additional analysis using only the records with complete demographic information. If the results align with those from the full dataset, it would suggest the missing data is not significantly biased. Of course, this is merely a suggestion, and the feasibility is entirely at the authors' discretion given the discussion timeline.

---

> > > ### Author Response · Authors · 2025-11-25
> > > **Response to Reviewer**
> > >
> > > Dear Reviewer 5CPC,
> > >
> > > We appreciate your prompt response to our additional comment. Unfortunately, we may be unable to fulfill this new request, as it would require re-training all models to ensure a fair comparison. We would like to emphasize that including additional data with unknown demographic attributes should not be viewed as an artifact but rather as a supplement to an already diverse population. Prior work has shown that PPG signals correlate with age and BMI and can be effectively regressed [1–2], while race is not so prevalant in the existing signal due to normalization (amplitude normalization removes racial information because PPG light interacts differently with varying skin tones [3]). Therefore we believe there are alternative approaches based on clustering to obtain these missing metadata, but admit we cannot do much better than that. While we do not have access to individual-level demographic details, we do know the general geographic distribution of our data collection, which spans the USA, Brazil, Bangladesh, and South Korea (**Written on Page 20, Section C.2.**) which we hope enhance our diversity claims.
> > >
> > > Upon reviewing our discussion thus far, we would also like to note that the strengths you identified in your initial review such as
> > > > "High Practical Significance",
> > >
> > > > "Novel Interpretability Framework",
> > >
> > > > "Experimental Rigor and Breadth",
> > >
> > > >"Strong Inductive Bias")
> > >
> > > These contributions represent the core contributions of this work. In addition to addressing all previously raised concerns, we have provided our codebase, improved visualizations, and clarified methodological details wherever necessary.
> > >
> > > **Given this, we would appreciate clarification on what specific limitations remain in our work beyond the demographic concerns. We are uncertain whether the current score assessment fully reflects the extent of our revisions and efforts.** **If there are additional weaknesses or missing components, we will continue to address them to the best of our ability. However, we are somewhat disheartened by the relatively harsh score our paper has received, despite having diligently responded to nearly all your concerns.** We thank you for your feedback but would like to ask for some clarity from the reviewer
> > >
> > > ## References:
> > >
> > > [1] Miller, Andrew C., et al. "A wearable-based aging clock associates with disease and behavior." Nature communications 16.1 (2025): 9264.
> > >
> > > [2] Abbaspourazad, Salar, et al. "Large-scale training of foundation models for wearable biosignals." arXiv preprint arXiv:2312.05409 (2023).
> > >
> > > [3] Bermond, Matteo, et al. "Reducing racial bias in SpO 2 estimation: The effects of skin pigmentation." 2023 45th Annual International Conference of the IEEE Engineering in Medicine & Biology Society (EMBC). IEEE, 2023.

---

> > > > ### Author Response · Authors · 2025-11-25
> > > > **Further comments related to data bias**
> > > >
> > > > Dear Reviewer,
> > > >
> > > > We'd like to make an additional comment related to the concerns raised about data biases on top of the ones we presented last night.
> > > >
> > > > We would like to mention that because we pre-trained nearly all our baseline comparisons (minus Chronos, and the open-weights PaPaGei) on the same exact data, that this again is the most fair comparison. For foundation model baselines like LSM (closed source) and PaPaGei, we ensured to train these models on the same data with their reported protocols because we wanted to have some clear apples-to-apples comparison regardless of what was reported in their paper.
> > > >
> > > > However to hopefully alleviate the reviewer, we have additionally trained a KNN model on the labeled components averaging PPG segments from the same user and then running inference on the unlabeled (NaN) demographics. This is our updated distribution though again we have no further way to verify these aside from knowing the general region:
> > > >
> > > > | **Category** | **Subgroup** | **N** | **% of total** |
> > > > |---------------|--------------|-------:|---------------:|
> > > > | **Sex** | Male | 36,990 | 77.6 |
> > > > |  | Female | 10,532 | 22.1 |
> > > > |  | Another gender identity | 122 | 0.3 |
> > > > | **Age** | 18–29 | 12,019 | 25.2 |
> > > > |  | 30–49 | 27,207 | 57.2 |
> > > > |  | 50–64 | 7,067 | 14.8 |
> > > > |  | 65+ | 1,351 | 2.8 |
> > > > | **Race** | White | 31,029 | 65.2 |
> > > > |  | Asian or Pacific Islander | 7,630 | 16.0 |
> > > > |  | Black or African American | 3,414 | 7.2 |
> > > > |  | American Indian or Alaskan Native | 592 | 1.2 |
> > > > |  | Another race | 4,979 | 10.4 |
> > > > | **BMI** | Underweight (<18.5) | 823 | 1.7 |
> > > > |  | Normal weight (18.5–24.9) | 13,626 | 28.5 |
> > > > |  | Overweight (25–29.9) | 16,634 | 34.9 |
> > > > |  | Obese I (30–34.9) | 8,745 | 18.5 |
> > > > |  | Obese II and III (≥35) | 7,816 | 16.4 |
> > > >
> > > > Again please let us know if we can further satisfy the reviewer with any concerns.

---

### Author Response · Authors · 2025-11-23
**We'd love to discuss**

Dear reviewers,

**We want to take the time again to thank the reviewers for their helpful constructive criticism they shared of our work which has drastically improved our maunscript.**

With roughly 10 days remaining in the author-reviewer discussion period, we wanted to ask if you could engage or acknowledge that you have read our rebuttal. **We did our best to run primarily all the additional experiments requested by individual reviewers** unless otherwise commented on, and **we believe that doing so warrants a score raise**. **If you are still not convinced by our methodology, we are happy to use the remaining time to address any further concerns or questions**.

The following changes were made since our submission:

- **All point by point responses are written** and contained on this **open review discussion**
- **There is a new version of the manuscript** where all tracked changed are **highlighted in blue text**

Thank you again as we know that many reviewers are also addressing their own rebuttals for their respective submissions.

---

### Author Response · Authors · 2025-11-28
**Summary of Review/Discussion for new Area Chair**

In light of the recent notice that the discussion period is closed (and scores were reverted), **we provide a summary of the reviews and discussion so the new area chair has full transparency independent of the scores**. **We thank all reviewers for their constructive feedback**, which substantially improved the submission. All changes are marked in blue in the revised PDF. **Our manuscript was fortunate to receive discussion prior to closure, and by its end all reviewers raised their scores relative to the initial evaluations.**

> **Reviewer 5CPC**: "Thank you to the authors for their efforts in addressing my concerns, which has resolved most of them. I have therefore raised my score."

> **Reviewer CnvE**: "Thanks for the rebuttal. Most of my concerns have been solved. I raised my score to 6."

> **Reviewer uicj**: "I thank the authors for additional clarifications ... The new evidence for Swin-T provides stronger evidence for HiMAE's potential on-device performance; thus, I recommend acceptance."

---

# Strengths

Some strengths noted consistently across reviewers include:
- **the on-device benchmarking and deployment** of SSL and wearable-based foundation models enabling efficient, on-device inference (5CPC, uicj)
- **the comprehensive and rigorous evaluation** of the proposed methodology (5CPC, CnvE, uicj)
- **the novel interpretability framework** that claims that different downstream tasks align with different temporal scales (5CPC, uicj)
- the alignment between the proposed convolutional inductive bias and the empirical results (5CPC).

---

# Weaknesses (that were mostly addressed)

We also provide some concerns raised by reviewers which we primarily address. These changes can be interpreted as what changes have been made during the discussion/revision period.

> Poor Presentation and Figure Quality (Rev. 5CPC, uicj)

We have revamped our manuscript with figures and tables that are much more easily legible whether read online or in print, while using a color pallette that adheres and can fully distinguish what constitutes as the best performance

> Reproducibility and Open Source  (Rev. 5CPC, uicj)

We recognize that reproducibility and open sourcing are essential for transparency. Although the pre-training data is proprietary and cannot be shared due to participant consent, we have provided the full model architecture (**p. 21**), hyperparameters (**pp. 28–29**), intermediate-layer probing (**p. 30**), and the on-device throughput and latency code blocks that initially caused confusion (**pp. 45–46**). We also include a [minimal codebase](https://anonymous.4open.science/r/HiMAE_ICLR_Synth-00E3/README.md) with the pre-training script and a runnable notebook using synthetic participant data for PVC arrhythmia classification.

> Novelty (Rev. CnvE)

We addressed the novelty concern along two dimensions.

First, we showed empirically that downstream tasks (e.g., arrhythmia detection, lab value prediction, sleep staging, and blood pressure) estimation depend on features emerging at different temporal scales. Our self-supervised architecture is designed to capture various morphology-level and trend-level structure, and the results support this. In the rebuttal to reviewer uicj, we clarified that transformers do not exhibit this behavior when probing intermediate layers, as each layer applies the same global attention block rather than learning scale-specific representations.

Second, we emphasized the on-device contribution. Many SSL and foundation-model approaches for wearables optimize for research benchmarks rather than deployability. By leveraging an inductive bias aligned with PPG signals, we show that compact models can match or surpass larger ones across diverse tasks, enabling practical on-device inference and supporting genuine real-world utility.

> More baselines (Rev. CnvE, uicj)

Two reviewers asked for additional baselines, and we largely met this request by training PaPaGei on our own pre-training data (formally only evaluated open-weight PaPaGei) and evaluating the general time-series foundation model Chronos on our tasks. Reviewer uicj suggested benchmarking LSM-2, but we noted a mismatch: LSM-2 is designed for settings with structured missingness, whereas our preprocessing removes segments with missing data. After evaluating the added baselines, HiMAE still outperforms competing models on most downstream tasks. We attribute the advantage of our smaller models to architectural choices and inductive biases that align closely with the structure of PPG signals, which exhibit quasi-periodicity and short-term non-stationarity.

> Modifying manuscript for clarity (Rev. 5CPC, CnvE, uicj) and adding a clinical interpretation section (Rev. uicj)

We have made substantial efforts to increase transparency in the manuscript, adding roughly eight pages of new content beyond the original submission. The text in blue highlights experimental protocols, motivations, and clinical interpretation.

---

> ### Author Response · Authors · 2025-11-28
> **Summary of Review/Discussion for new Area Chair (Continued)**
>
> ### Closing remarks
>
> In addition to the weaknesses highlighted in the summary, we would like to acknowledge that there were more minor questions/concerns regarding our work. The additional questions and minor weaknesses raised by the reviewers were, for the most part, resolved through further experimentation or detailed clarification, and the area chair is of course welcome to read or skim the remainder of the exchange for full context.
>
> **We feel genuinely fortunate that our reviewers were active, thoughtful, and deeply engaged before the discussion period closed. Their improved sentiment toward the work emerged organically through that dialogue, and we hope it is similarly evident and persuasive to the Area chair. Their feedback substantially strengthened the manuscript and materially improved its transparency.**
>
> **We also want to extend our sincere thanks to the area chair in advance, who will be working under considerable time over the coming weeks to make difficult decisions across many submissions. With that in mind, we prepared this summary to streamline your evaluation, especially given how much discussion our work received.**
>
> **The most compelling part of this review and discussion phase was recognizing that every concern and question raised by the reviewers was addressable, and that we invested substantial effort in resolving them thoroughly. Regardless of either the initial scores or their updates after rebuttal, we believe we have taken the necessary steps to clear up sources of confusion or uncertainty in the manuscript, and at this point of the revision, we feel we have addressed all their primary concerns while being as transparent as possible.**
>
> Thank you all again for your time and effort for making this conference possible.

---

### Meta-Review · Area_Chair_SyQ3 · 2026-01-07

**Summary:**

The submission presents a novel and useful framing ("resolution hypothesis") and a model (HiMAE) that achieves strong downstream performance while enabling true on-device inference. The authors substantially improved the manuscript during rebuttal: they expanded experiments, added important baselines (including PaPaGei variants and Chronos), improved interpretability analyses and visualizations, clarified on-device benchmarking protocols, and provided statistical uncertainty estimates.

After the authors’ revisions, the reviewers raised their scores (derived from the discussions) and two recommend acceptance. Remaining concerns primarily relate to reproducibility and dataset-demographic transparency rather than core scientific validity. Because the scientific contribution is convincing and reviewers converged positively after revisions, I recommend acceptance provided the authors meet the conditions below, given that reproducibility is a major concern.

The authors must provide the following artifacts and clarifications at camera-ready. These are strongly recommended conditions for publication:

1) Artifact release
Publish the minimal codebase and runnable notebook referenced in the rebuttal (the anonymized pretraining/evaluation scripts and the synthetic PVC notebook). The artifact must be accessible via a stable URL (e.g., a public GitHub repository or the conference supplementary material system) and include clear README instructions for reproducing the linear-probe and on-device benchmarking experiments using the synthetic data.
If any artifact cannot be publicly shared for legal reasons, provide an explicit statement explaining the legal constraint.

2) Synthetic dataset
Provide the synthetic dataset used in the runnable notebook (or a clearly scripted generator that reproducibly creates it), plus a short description of how the synthetic data was generated, its statistical properties, and how it relates to the real dataset. This enables reviewers and readers to run the supplied code and validate the pipeline without access to proprietary data.

**Reviewer Concerns:**

Through the extensive discussions, all the reviewers' concerns seem to have been addressed, except for reproducibility, which is a major concern for ICLR.

In a nutshell, the work is empirically strong but not fully reproducible because of proprietary pretraining data & unavailable weights.

The PC chairs can require that the minimal code and synthetic notebook be published at camera-ready and that the authors provide a short, precise legal statement about licensing and on sharing derivatives.

**Reviewer Scores:**

The reviewers have stated in the responses that they have all incresed their scores and/or recommend acceptance.

---

### Decision · Program_Chairs · 2026-01-26

Accept (Poster)